# Spatial heterogeneity of GHG dynamics across an estuarine ecosystem

Nicolas-Xavier Geilfus<sup>1</sup>, Bruno Delille<sup>2</sup>, Anna Villnäs<sup>1</sup>, Alf Norkko<sup>1</sup>

- <sup>1</sup>Tvärminne Zoological Station, University of Helsinki, Finland
- <sup>2</sup>Chemical Oceanography Unit, Université de Liège, Belgium
- 5 Correspondence to: Nicolas-Xavier Geilfus (nicolas-xavier.geilfus@helsinki.fi)

# Abstract.

Coastal ecosystems are critical components of the global carbon cycle, exerting a disproportionate influence on the carbon budget despite their limited spatial extent. Estuaries remain understudied despite being dynamic sources of the three most potent greenhouse gases (GHGs): carbon dioxide (CO<sub>2</sub>), methane (CH<sub>4</sub>), and nitrous oxide (N<sub>2</sub>O). Such shallow coastal ecosystems are highly heterogeneous, shaped by strong physical, biogeochemical, and biological gradients. Combined with spatial variability in coastal biodiversity, these gradients strongly shape carbon cycling at both local and global scales. However, large uncertainties persist due to limited and spatially patchy measurements, highlighting the need for improved constraints on GHG budgets and their sensitivity to biodiversity loss and climate change.

We measured surface seawater partial pressure of CO<sub>2</sub> (pCO<sub>2</sub>), CH<sub>4</sub>, and N<sub>2</sub>O concentrations, along with seawater physical and biogeochemical properties, and air-sea gas exchange, at 21 sites in southwest Finland (Baltic Sea). Sampling followed a transition from estuarine inner bays to the outer archipelago, covering diverse soft-sediment habitats, from sheltered to exposed areas, across a salinity gradient. Surface water pCO<sub>2</sub> and N<sub>2</sub>O concentration ranged from undersaturated (160 ppm and 9 nmol L<sup>-1</sup>, respectively) to supersaturated (2521 ppm and 25 nmol L<sup>-1</sup>, respectively), compared to the atmosphere, resulting in an uptake of -36 and -0.0021 mmol m<sup>-2</sup> d<sup>-1</sup>, and a release up to 220 and 0.0383 mmol m<sup>-2</sup> d<sup>-1</sup>, respectively. CH<sub>4</sub> concentrations were consistently supersaturated (19 to 469 nmol L<sup>-1</sup>) compared to the atmosphere, resulting in a net source to the atmosphere from 0.014 to 1.39 mmol m<sup>-2</sup> d<sup>-1</sup>.

Freshwater input from the Karjaanjoki River and its mixing with seawater mainly determined the overall spatial patterns of GHGs. However, deviations from this salinity-driven control were observed. In sheltered sites within the archipelago, elevated pCO<sub>2</sub>, CH<sub>4</sub>, and N<sub>2</sub>O concentrations likely reflected benthic processes, including enhanced organic matter respiration and methanogenesis in warm, late-summer shallow waters, where limited oxidation favoured CH<sub>4</sub> accumulation. At exposed sites, mixing processes had a stronger control, resulting in lower GHG concentrations. Our results show that both physical mixing and benthic processes influence coastal GHG dynamics, with benthic ecosystems playing a key but still poorly constrained role. Air—sea GHG exchanges were dominated by CO<sub>2</sub>, while CH<sub>4</sub> and N<sub>2</sub>O contributed differently as a source and a sink. The balance between production and consumption processes, particularly within benthic habitats, is therefore critical for understanding coastal contributions to the global carbon budget.

# 1. Introduction

Coastal ecosystems are increasingly recognized as critical components of the global carbon cycle (Bauer et al., 2013). Although covering only about 7% of the oceanic area, they play a disproportionately large role in biogeochemical cycling due to their location at the land—sea interface (Gattuso et al., 1998; Wollast, 1998). These environments act as hubs of exchange across biomes (Resplandy et al., 2024), contributing to the global oceanic uptake of anthropogenic carbon by absorbing carbon dioxide (CO<sub>2</sub>) from the atmosphere and either burying, transforming, or releasing the carbon that land ecosystems deliver to coastal waters (Regnier et al., 2022). Due to intense inputs of nutrients and carbon from land and open ocean at continental margins, combined with high rates of biological production and degradation, coastal waters are among the most biogeochemically active regions of the biosphere (Wollast, 1998). Shallow coastal waters such as mangroves, salt marshes, and seagrass meadows effectively sequester large amounts of atmospheric carbon through vegetation growth and long-term sediment accumulation, making significant contributions to long-term carbon storage (Lovelock and Duarte, 2019; Macreadie et al., 2021; Mcleod et al., 2011). These so-called "blue carbon" ecosystems have been proposed as nature-based solutions for short-term climate change mitigation (Hoegh-Guldberg et al., 2019; Lovelock and Duarte, 2019).

Vegetated blue carbon ecosystems exemplify the value of healthy habitats in long-term carbon storage. Still, they cover a small part of the coastal oceans (thebluecarboninitiative.org), and measurements from other key habitats remain scarce and largely neglected in carbon cycling budgets (James et al., 2024). Estuarine areas are characterized by strong physical and chemical gradients and host transitional biodiversity that bridges terrestrial and marine ecosystems (Elliott and Whitfield, 2011) and are recognized as hot spots for carbon cycling and GHG exchange with the atmosphere (Borges et al., 2005; 2016; Cai et al., 2013; Frankignoulle et al., 1998; Humborg et al., 2019; Rosentreter et al., 2021). The global oceans absorb roughly 25% of anthropogenic CO<sub>2</sub> emissions annually (Watson et al., 2020). While coastal waters contribute to this sink by taking up atmospheric CO<sub>2</sub> (Chen and Borges, 2009; Regnier et al., 2022), many nearshore systems like estuaries have been recognized as net heterotrophic ecosystems with negative net ecosystem production (Gattuso et al., 1998; Testa et al., 2012), leading to the production and release of both CO<sub>2</sub> and methane (CH<sub>4</sub>) (Abril and Borges, 2005; Bonaglia et al., 2025; Borges et al., 2018). Despite their small size, emissions from near-shore ecosystems such as estuaries could nearly balance the carbon uptake by marginal seas (Borges et al., 2005; Chen and Borges, 2009). Global CO<sub>2</sub> emissions from estuaries range from 0.1 to 0.6 PgC per year (Borges et al., 2018), an amount equivalent to 5-30% of the oceanic CO<sub>2</sub> sink of ~ 2 PgC per year (Le Quéré et al., 2016), while CH<sub>4</sub> emissions range from 1 to 7 Tg per year, mainly driven by organic matter accumulation, anaerobic decomposition in sediments, and production in submerged plants (Borges et al., 2018; Rosentreter et al., 2018). However, CH4 emissions are probably underestimated due to difficulties in accounting for ebullition and gas flaring (Borges et al., 2016; Humborg et al., 2019). The complexity of these pathways, combined with a lack of systematic, high-resolution, and long-term measurements, hampers robust global assessments of coastal CH<sub>4</sub> emissions (Roth et al., 2022). Besides CO<sub>2</sub> and CH<sub>4</sub>, coastal ecosystems are important but poorly constrained sources of nitrous oxide (N<sub>2</sub>O), a potent greenhouse gas with a global warming potential 273 times higher than CO<sub>2</sub> over 100 years (IPCC 2023). N<sub>2</sub>O in oceanic environments is formed as a byproduct

during nitrification and as an intermediate during denitrification, both of which depend strongly on oxygen concentration and are microbial processes that occur in the water column, in sediments, and within suspended particles (Bange, 2006). Coastal ecosystems are recognized as significant sources of N<sub>2</sub>O to the atmosphere (Bange, 2006; Cheung et al., 2025; Resplandy et al., 2024). However, coastal estimates remain uncertain due to sparse measurements and high spatial heterogeneity. Given the strong warming potential of N<sub>2</sub>O, understanding its fluxes from estuaries is essential for a complete picture of coastal greenhouse gas dynamics and determining how these emissions, along with those of CH<sub>4</sub> (with a warming potential of 27; IPCC 2023), offset the coastal uptake of CO<sub>2</sub> (Resplandy et al., 2024; Rosentreter et al., 2018; Roth et al., 2023). This questions the efficiency of blue carbon ecosystems in mitigating the rise of GHG concentrations in the atmosphere and their related radiative effects (Kristensen et al., 2025; William and Gattuso, 2022).

This study investigates the dynamics of CO<sub>2</sub>, CH<sub>4</sub>, and N<sub>2</sub>O (GHGs) along a salinity gradient and across contrasting coastal habitats within an estuary. We combined detailed field measurements of surface seawater physical and biogeochemical properties with both *in situ* measurements and calculated estimates of air—sea GHG exchange. This approach allowed us to capture spatial variability, identify the underlying biogeochemical drivers, and examine interactions with coastal habitats. In doing so, we aim to disentangle the biogeochemical controls of GHG dynamics in estuarine environments and contribute to improving global estimates of the coastal ocean's GHG emissions.

#### 2. Materials and methods

# 80 2.1 Study area

This study was conducted within a 30 km radius of the Tvärminne Zoological Station (TZS), located at the southwestern tip of mainland Finland. The study area encompasses the coastal waters extending from the cities of Hanko, Tammisaari, and Pohja (Fig. 1), where the coastal environment is characterized by strong spatial heterogeneity due to the intricate bathymetry and geomorphology of the archipelago system (Asmala and Scheinin, 2024) and enhanced human impact. The studied area receives substantial freshwater from the Karjaanjoki river (in Pohja, runoff 0.59 km³ yr⁻¹, Räike et al., 2012), delivering relatively large amounts of allochthonous carbon and nutrients to the system, resulting in eutrophication and elevated concentrations of organic carbon in the water column across the inner archipelago area (Fleming-Lehtinen et al., 2015). Sampling sites (N = 21) were selected to encompass a wide range of soft-sediment habitats and to represent a spatial gradient (50 km) from the outer to innermost archipelago, encompassing exposed to sheltered bays. Exposed sites are characterized by a higher degree of exposure to wave, wind, current energy, and greater water exchange (sites 1-4 and 14-21), while sheltered sites are more enclosed and experience more limited water circulation (sites 5-13). Exposed sites in the outer archipelago (between Hanko and TZS, sites n°14-21, Fig. 1) are characterized by sandy sediments (96.2–100% sand) with low organic matter content (0.3-1.3%). These sites host both marine and freshwater plant species (Gustafsson and Norkko, 2019; Lammerant et al., 2025), and their faunal biomass is dominated by bivalves and polychaetes (Gammal et al., 2019; Mäkelin et al., 2024). Sheltered sites within the archipelago (sites n°5-13) are characterized by sediments containing high proportions of

clay/silt (20.8-64.3%) and organic matter (2.4-13.2%; Lammerant et al., 2025). At these sites, fast-growing fresh- and brackish-water macrophytes are often found, while faunal biomass is dominated by gastropods (Mäkelin et al., 2024). In Pojoviken Bay, a 14 km inlet between Tammisaari and Pohja (sites n°1-4), dense vegetation occurs down to ~2 m depth but becomes sparse at greater depths (>3 m) due to light limitation (Lammerant et al., 2025). Deeper sites are characterized by high organic matter content (>13%) in the sediment and low faunal biomass (Gammal et al., 2019). Sampling was conducted from 14 August to 18 September 2023.

# 2.1 Sampling procedure

Physical and biogeochemical properties of surface seawater, including the partial pressure of dissolved CO<sub>2</sub> (pCO<sub>2</sub>) and CH<sub>4</sub> concentration, were measured *in situ* using a custom-built flow-through system. Seawater, from a depth of 50 cm, was pumped onboard using a submersible pump and directed into two separate flow-through systems.

The first water flow (3 L min<sup>-1</sup>) was dedicated to gas extraction using a double showerhead equilibrator (Sunburst) equipped with a temperature probe (Hex fitting thermistor, ThermX, 0.1°C accuracy) and a barometric pressure sensor (BARO-A-4V, All Sensors). From the equilibrator, a continuous air flow (2 L min<sup>-1</sup>) circulated in a closed loop to an infrared gas analyser (IRGA, LI-COR, LI-7810). The analyser measured the dry mole fractions of CO<sub>2</sub> and CH<sub>4</sub>, denoted as xCO<sub>2</sub> and xCH<sub>4</sub>, respectively, as the gas passed through a Peltier cooler to remove the excess water before entering the IRGA. Both gas concentration and temperature were recorded every second until an equilibrium was reached (up to 45 minutes). The measured xCO<sub>2</sub> was later converted into pCO<sub>2</sub> and corrected for *in situ* temperature using the Matlab CO2SYS v3 (Sharp et al., 2023). Surface seawater CH<sub>4</sub> concentration (in nmol L<sup>-1</sup>) was computed from xCH<sub>4</sub>, *in situ* temperature, salinity, and the solubility coefficients from Wiesenburg and Guinasso (1979).

The second water flow (2 L min<sup>-1</sup>) was directed through a thermosalinograph (SeaBird, TSG045), a fluorometer for chlorophyll *a* (chl-*a*), turbidity, and phycocyanin concentrations (Chelsea, TriLux), a fluorometer for coloured dissolved organic matter (CDOM, Chelsea, UviLux), and an optode for dissolved oxygen concentration and saturation (Aanderaa 4531). Each sensor was factory-calibrated before deployment, and data were logged at 1-second intervals until equilibrium for xCO<sub>2</sub> and xCH<sub>4</sub> was reached (up to 45 min). A 5-second average was then recorded.

Discrete surface water samples were collected for total alkalinity (TA), dissolved inorganic carbon (DIC), methane (CH<sub>4</sub>), and nitrous oxide (N<sub>2</sub>O) concentration using a peristaltic pump (Cole Palmer, Masterflex Environmental Sampler) equipped with Tygon tubing. Seawater samples were collected into 12 mL gas-tight vials (Exetainers, Labco High Wycombe, UK) for DIC, 60 mL borosilicate vials for TA, and 60 mL serum bottles for dissolved CH<sub>4</sub> and N<sub>2</sub>O concentrations. Samples were preserved by adding 1% of the sample volume of a saturated mercuric chloride (HgCl<sub>2</sub>) solution. Samples were stored in the dark at room temperature until analysis.

Air-sea exchanges of CO<sub>2</sub> and CH<sub>4</sub> were measured using the accumulation chamber technique (Frankignoulle, 1988). The chamber consists of a polyethylene container (internal diameter: 34 cm, height: 14 cm, total volume = 11.4 L) connected in a closed loop to the IRGA. The air partial pressure of CO<sub>2</sub> (pCO<sub>2</sub>) and CH<sub>4</sub> (pCH<sub>4</sub>) within the chamber was recorded every

130

second for 10 minutes. The flux was computed from the slope of the linear regression of pCO<sub>2</sub> and pCH<sub>4</sub> against time ( $R^2 > 0.99$ ) following the method of Frankignoulle (1988), accounting for the air volume enclosed within the chamber. The uncertainty of the flux calculation, based on the standard error on the regression slope, was approximately 3% on average. Direct measurements of N<sub>2</sub>O air-sea exchanges were not possible during our survey. Therefore, the air-sea exchange of N<sub>2</sub>O, as well as CO<sub>2</sub> and CH<sub>4</sub> (i.e., F<sub>X</sub>, where X represents N<sub>2</sub>O, CO<sub>2</sub>, or CH<sub>4</sub>, respectively) was parameterized using the bulk formula:

$$F_X = kK_{0(X)} (pX_{(sw)} - pX_{(atm)}) \tag{1}$$

where k (m s<sup>-1</sup>) is the gas transfer velocity,  $K_0$  (mol m<sup>-3</sup> atm<sup>-1</sup>) is the temperature and salinity dependent gas solubility determined from Weiss (1974) for CO<sub>2</sub>, Wiesenburg and Guinasso (1979) for CH<sub>4</sub>, and Weiss and Price (1980) for N<sub>2</sub>O, and  $pX_{(sw)}$  and  $pX_{(atm)}$  are the measured partial pressure of gas in the surface seawater and the air, respectively.

This approach strongly relies on selecting an appropriate parameterization for the gas transfer velocity, k. Numerous theoretical, laboratory, and field studies established that k depends on a variety of parameters, including capillary and breaking waves, boundary layer stability, air bubbles, surfactant surface films, evaporation and condensation, precipitation, water currents (including tides), as well as turbulence at the air-water interface (Borges et al., 2004). The parameterization of k is most often expressed as a function of wind speed. This relationship is well established for the open ocean, where wind stress is the main source of turbulence (e.g., Wanninkhof, 2014). In estuarine environments, significant regression functions between k and wind speed have also been reported (Borges et al., 2004; Raymond et al., 2000). However, such formulations have been proven to be site-specific and may not fully account for other controlling processes, such as turbulence at the air-water interface (Borges et al., 2004). Since no k-wind relationship has been developed for our study area, we adopted the formulation proposed for the Randers fjord by Borges et al. (2004), as its physical characteristics, particularly tidal amplitude and freshwater discharge, are most comparable to those of our site:

$$k = 1.2 + 2.3u \left(\frac{Sc_{(balt)}}{660}\right)^{-0.5} \tag{2}$$

where u is the wind speed (m s<sup>-1</sup>) and  $Sc_{(balt)}$  is the Schmidt number at the sampling site, based on local temperature, salinity, and gas molecules.  $Sc_{(balt)}$  is extrapolated from freshwater and seawater (S = 35) coefficients from Wanninkhof (2014) for the salinity observed at the sampling site (Roth et al., 2023).

Wind speed (in m s<sup>-1</sup>, METEK, uSonic-3 Scientific) and air temperature (in °C, Vaisala, HMP155) are measured at the 3.2 and 2 m height above sea level at the newly established Integrated Carbon Observation System (ICOS) coastal site at TZS (ICOS code FI-Tvm; Fig. 1). Daily averaged wind speed, extrapolated to a height of 10m using the relationship from Hsu et al., (1994), assuming a near-neutral atmospheric stability conditions, was used to calculate the air-sea exchanges of GHGs. Atmospheric concentrations of CO<sub>2</sub> and CH<sub>4</sub> were measured at each site before the air-sea flux measurements. For N<sub>2</sub>O, we used the monthly average atmospheric concentration of 0.3367 ppm, measured at Pallas-Sammaltunturi, GAW Station, Finland (PAL, https://gml.noaa.gov/dv/iadv/graph.php?code=PAL&program=ccgg&type=ts).

The recorded data on the IRGA for both seawater and air-sea flux measurements were filtered by removing measurements taken during the transition period between stations and ambient air sampling, as the IRGA requires time to respond to sharp concentration changes. Additionally, data affected by improper functioning (i.e., seawater flow < 1.5 L min<sup>-1</sup>) were discarded.

# 2.3 Sample analysis

TA was determined by Gran titration (Gran, 1952) using an AS-ALK2 titration system (Apollo SciTech), where a 15-ml sample was titrated with a standard 0.1 M HCl solution. DIC was measured on a DIC analyser (Apollo SciTech) by acidification of a 0.75 ml subsample with 1 ml 10% H<sub>3</sub>PO<sub>4</sub>, and quantification of the released CO<sub>2</sub> with a non-dispersive infrared CO<sub>2</sub> analyser (LI-COR, LI-7000). Results were then converted from μmol L<sup>-1</sup> to μmol kg<sup>-1</sup> based on sample density, calculated from salinity and temperature at the time of the analysis. Accuracies of ±3 and ±2 μmol kg<sup>-1</sup> were determined for TA and DIC, respectively, from routine analysis of certified reference materials (A.G. Dickson, Scripps Institution of Oceanography, San Diego, CA, USA).

Discrete seawater samples for CH<sub>4</sub> and N<sub>2</sub>O concentrations were measured via the headspace equilibrium technique (25 ml N<sub>2</sub> headspace in 60 ml serum bottles) and measured with a gas chromatograph (SRI 8610C) with flame ionization detection, and electron capture detection calibrated with CH<sub>4</sub>: CO<sub>2</sub>:N<sub>2</sub>O: N<sub>2</sub> mixtures (Air Liquide) of 1.0, 10, and 30 ppm CH<sub>4</sub> and 0.2, 1.0, and 6.0 ppm N<sub>2</sub>O. The measurement precision was 8 % for CH<sub>4</sub> and 4 % for N<sub>2</sub>O.

### 175 **3. Results**

# 3.1 Atmospheric forcing and seawater physical properties

During the survey period, atmospheric temperatures remained relatively stable, averaging  $18.0^{\circ}$ C (SD = 1.5, n = 1989; Fig. 1a). A short warm event occurred on the  $7^{th}$  of August, when the air temperature peaked at  $26.3^{\circ}$ C. In contrast, short cold spells were observed on 5 and 7 September, with temperatures dropping to 13.7 and  $11.5^{\circ}$ C, respectively. With relatively stable atmospheric temperature, surface seawater temperature (SST) showed limited variation, ranging from 17.2 to  $19.5^{\circ}$ C (mean =  $18.4^{\circ}$ C, SD = 0.66, n = 21; Fig. 1c). The Karjaanjoki River, being the main source of freshwater input to the area, created a pronounced salinity gradient, from nearly fresh surface seawater salinity (SSS = 0.07) at the river mouth (in Pohja) to a SSS of 6.36 in Hanko (Fig. 1d).

# 3.2 Seawater biogeochemical properties

Surface seawater chlorophyll-*a* (chl-*a*) concentration ranged from 3.7 to 37.7 μg L<sup>-1</sup> (mean = 14.4, SD = 8.1, n = 21, Fig. 1e), with the highest concentration observed in the sheltered bay around Tammisaari (S5 and S6). Turbidity ranged from 5.0 to 8.4 FTU (mean = 6.1, SD = 0.8, n = 21, Fig. 1f), with the highest values observed in sheltered bays within the archipelago (S8, S9, and S11). Coloured dissolved organic carbon (CDOM) ranged from 3.0 to 10.6 μg L<sup>-1</sup> (mean = 5.9, SD = 2.6, n = 21, Fig. 1g) and exhibited a salinity-driven gradient, with the highest concentrations observed towards Pohja and the lowest towards

Hanko. However, one site in the archipelago (S8) and near Hanko (S19) exhibited elevated CDOM concentrations compared to other locations in their respective area, reaching up to 10.0 and 10.6, respectively. Dissolved oxygen (DO) saturation levels ranged from 74.3 to 112.7 % (mean = 94.1, SD = 9.6, n = 21, Fig. 1h), with most of the sites undersaturated in O<sub>2</sub>, excepted for two sites closed by Tammisaari (S3 and S4) and four sites in closed vicinity of TZS (S14, S15, S18, and S19).

Surface water pCO<sub>2</sub> ranged from strong undersaturation (160 ppm) to a significant supersaturation relative to the atmosphere (average atmospheric concentration measured during this survey = 406 ppm) at S2 (1702 ppm) and up to 2521 ppm at the river mouth (triangle on Fig. 2a). Surface water CH<sub>4</sub> concentrations were consistently supersaturated (up to 14609 %, data not shown) compared to the atmosphere (average atmospheric concentration measured during this survey = 2.05 ppm), with concentrations ranging from 19 to 469 nmol L<sup>-1</sup> (Fig. 2b). Undersaturated pCO<sub>2</sub> (from 160 to 403 ppm) and low CH<sub>4</sub> concentrations (from 19 to 34 nmol L<sup>-1</sup>) were mostly observed at open sites between TZS and Hanko (S14 to S21), except for S17, where pCO<sub>2</sub> and CH<sub>4</sub> reached 485 ppm and 48 nmol L<sup>-1</sup>, respectively. Seawater pCO<sub>2</sub> and CH<sub>4</sub> concentrations increased across the archipelago and Pojoviken Bay, reaching their maximal values at the mouth of the Karjaanjoki River. Local lows were observed in Pojoviken Bay (S3 and S4), where pCO<sub>2</sub> was undersaturated (192 and 238 ppm, respectively), and CH<sub>4</sub> concentrations were relatively low (83 and 67 nmol L<sup>-1</sup>, respectively). N<sub>2</sub>O concentration ranged from 9 to 25 nmol L<sup>-1</sup>, with higher concentrations observed at the mouth of the Karjaanjoki River (Fig. 2c). South of Tammisaari, surface waters were either undersaturated in N<sub>2</sub>O (S8, S13, and S16) or close to the atmospheric equilibrium (S6 and S12, data not shown). TA and DIC ranged from 668 to 1631 and from 720 to 1550  $\mu$ mol kg<sup>-1</sup>, respectively, with minimum TA and DIC concentration measured at the mouth of the Karjaanjoki River. Both TA and DIC exhibited conservative behaviour with changes in salinity (Fig. 3a).

#### 3.3 Air-sea exchanges of greenhouse gases

The area presented contrasts in terms of sink and source of CO<sub>2</sub> and N<sub>2</sub>O, but was clearly a source of CH<sub>4</sub>. Measured air-sea exchanges of CO<sub>2</sub> ranged from -36 to 150 mmol m<sup>-2</sup> d<sup>-1</sup>, where negative values indicate net uptake of atmospheric CO<sub>2</sub> (Fig. 2d). In contrast, the study area acted as a net source of CH<sub>4</sub> to the atmosphere, with fluxes ranging from 0.014 to 1.39 mmol m<sup>-2</sup> d<sup>-1</sup> (Fig. 2e). Overall, the calculated fluxes showed a similar in term of pattern and range compared to the observed values, ranging from -26.4 to 220.3 mmol m<sup>-2</sup> d<sup>-1</sup> for CO<sub>2</sub> (Fig. 2f) and from 0.03 to 1.1 mmol m<sup>-2</sup> d<sup>-1</sup> for CH<sub>4</sub> (Fig. 2g). Calculated fluxes of N<sub>2</sub>O ranged from -0.0021 to 0.0383 mmol m<sup>-2</sup> d<sup>-1</sup>, with sites in the archipelago and towards TZS acting as a slight sink of N<sub>2</sub>O (Fig. 2h).

# 4. Discussion

Surface water pCO<sub>2</sub>, CH<sub>4</sub>, and N<sub>2</sub>O concentrations showed significant spatial variation, spanning several orders of magnitude (Fig. 2). Humborg et al. (2019) reported pCO<sub>2</sub> and CH<sub>4</sub> levels following the salinity gradient driven by the Karjaanjoki River, with values reaching up to 1583 μatm and 70 nmol L<sup>-1</sup>, respectively. However, these measurements were conducted along the

deeper part of the main channel of the archipelago, and they did not explore Pojoviken Bay, whereas our study focused on shallow (< 4m depth) nearshore ecosystems. Asmala and Scheinin (2024) observed a similar range of pCO<sub>2</sub> and CH<sub>4</sub> concentrations across a 2000 km<sup>2</sup> coastal region surrounding the Hanko Peninsula, with the large magnitude of both gases associated with excessive organic matter loads, elevated primary production, trapping and accumulation of allochthonous organic matter, and sedimentary conditions favourable to CH<sub>4</sub> production. Focusing on nearshore shallow (< 4m depth) coastal habitats around the island of Askö (Sweden, northern Baltic Sea), Roth et al. (2023) reported strong spatial variability in surface water pCO<sub>2</sub> and CH<sub>4</sub> concentrations and emphasized that their dynamics were strongly habitat specific. The N<sub>2</sub>O concentrations we observed were in the same range as those observed by Aalto et al. (2021) in the same area, with concentrations ranging from 25 to 50 nmol L<sup>-1</sup> within Pojoviken Bay and from 10 to 30 nmol L<sup>-1</sup> within the archipelago. Aalto et al. (2021) reported that higher N<sub>2</sub>O concentrations were associated with higher nitrate concentrations and inputs of allochthonous carbon, while lower N<sub>2</sub>O concentrations were associated with efficient internal recycling of N. The Kendall's τ coefficient has been calculated to investigate the correlation between surface water pCO<sub>2</sub>, CH<sub>4</sub>, and N<sub>2</sub>O concentration, as well as between physical and biogeochemical parameters and all three GHGs (Fig. 4). Surface pCO<sub>2</sub> and CH<sub>4</sub> concentrations show a moderate ( $\tau = 0.5636$ ) and statistically significant (p = 0.0165) positive relationship (Fig. 4a), suggesting that processes driving high pCO<sub>2</sub>, such as respiration and organic matter degradation, are likely also contributing to elevated CH<sub>4</sub> levels. N<sub>2</sub>O is not correlated with pCO<sub>2</sub> and shows a moderate correlation with CH<sub>4</sub> ( $\tau = 0.4182$ ), although this relationship is only marginally significant (p = 0.0866). Among all tested biogeochemical variables, SSS and O<sub>2</sub> saturation show strong and significant negative correlations with pCO<sub>2</sub> and CH<sub>4</sub> (Fig. 4b). N<sub>2</sub>O is only strongly negatively correlated with salinity (τ = -0.8182, p = 0.00013) and shows a marginal correlation with CDOM ( $\tau = 0.4545$ , p = 0.060). CDOM is positively correlated with CH<sub>4</sub> ( $\tau = 0.4952$ , p = 0.0013) while chl-a is moderately and significantly correlated with CH<sub>4</sub> ( $\tau = 0.4095$ , p = 0.009). SST and turbidity show weak and non-significant relationships with the three gases.

# 4.1 Drivers for variability in surface water pCO<sub>2</sub>, CH<sub>4</sub>, and N<sub>2</sub>O concentration

The Karjaanjoki river is the main source of freshwater into the studied area, delivering large amounts of allochthonous carbon and nutrients to the system (Fleming-Lehtinen et al., 2015). Changes in SSS have been suggested to strongly influence the concentration of all three GHGs (Fig. 4b), with terrestrial runoff likely contributing to elevated CH<sub>4</sub> concentrations in the area (Asmala and Scheinin, 2024). At a first glimpse, only N<sub>2</sub>O exhibited a strong relationship with salinity (R<sup>2</sup> = 0.95), while pCO<sub>2</sub> (R<sup>2</sup> = 0.58) and CH<sub>4</sub> (R<sup>2</sup> = 0.39) exhibited weaker correlations (Fig. 5, dotted black line). However, when focusing exclusively on exposed sites, the correlation with salinity strengthened considerably, reaching values up to 0.79 for pCO<sub>2</sub>, 0.94 for CH<sub>4</sub>, and 0.96 for N<sub>2</sub>O (Fig. 5, dashed red line). The impact of water mixing on surface-water pCO<sub>2</sub> can be assessed by estimating the variation in pCO<sub>2</sub> resulting only from the physical mixing of Karjaanjoki River water (S = 0, TA = 668  $\mu$ mol kg<sup>-1</sup>, DIC = 720  $\mu$ mol kg<sup>-1</sup>) with a seawater endmember (black symbol in Fig. 3, S = 6.42, TA = 1635  $\mu$ mol kg<sup>-1</sup>, DIC = 1586  $\mu$ mol kg<sup>-1</sup>). The seawater endmember was collected on 18 September 2023 at 59.77393 °N, 23.2607167 °E, about 7.5 km south of TZS. This calculation excludes the effects of biological activity, gas exchange, and potential precipitation or dissolution of calcium

carbonate. Assuming that changes in SSS reflect a mixing ratio between the Karjaanjoki River and the seawater endmembers and based on the observed linear relationships between salinity and both TA and DIC (Fig. 3a), representing conservative mixing between the two water masses, TA and DIC can be estimated based on salinity. The estimated TA and DIC are then used to compute pCO<sub>2</sub> (blue dashed line, Fig. 5a) using CO2SYS v3 (Sharp et al., 2023), applying the carbonic acid dissociation constants (K<sub>1</sub> and K<sub>2</sub>) of Millero et al. (2006) and the KHSO<sub>4</sub> formulation of Dickson (1990). While TA and DIC behaved conservatively during mixing, pCO<sub>2</sub> showed a pronounced nonlinear response, similar to the one observed for exposed sites (Fig. 5a). Such behaviour has already been reported in estuaries (Abril et al., 2021; Cai et al., 2013). This suggests that the input of CO<sub>2</sub>, CH<sub>4</sub>, and N<sub>2</sub>O supersaturated water from the Karkaanjoki River is a major source for all three GHGs across the study area, with dilution playing a dominant role in GHG dynamics at exposed sites. While changes in SSS appear to explain most of the variability observed in pCO<sub>2</sub>, CH<sub>4</sub>, and N<sub>2</sub>O concentrations, deviations from the expected salinity-driven pattern, such as those observed within the archipelago, at sheltered sites, likely reflect additional local processes.

Many processes affecting the carbonate system are best described by examining the associated changes in DIC and TA (Zeebe and Wolf-Gladrow, 2001). However, most of the changes in TA and DIC in surface seawater appeared to be driven by salinity changes, with estimated endmember (S=0) values of 662 and 712 µmol kg<sup>-1</sup> for TA and DIC, respectively (Fig. 3a). These estimates closely match the measured TA and DIC at the mouth of the Karjaanjoki River (668 and 720 µmol kg<sup>-1</sup>, respectively). Therefore, to discard the impact of freshwater inputs, TA and DIC were normalized to the average surface seawater salinity of 5 (denoted as nTA and nDIC, respectively) using the normalization of Friis et al. (2003). In Figures 3b, c, and d, the dotted lines illustrate the expected responses of TA and DIC to different biogeochemical processes. Biological activity (photosynthesis/respiration) affects both TA and DIC in a ratio of -0.16 (Lazar and Loya, 1991), air-sea exchange of CO<sub>2</sub> only affects DIC, and the precipitation-dissolution of calcium carbonate affects TA and DIC in a 2:1 ratio. However, the effect of calcifying primary producers in the carbon pool can be neglected in the Baltic Sea, except in the benthic zone (Tyrrell et al., 2008). The sampling sites are compared to the seawater endmember (S = 6.42,  $TA = 1635 \mu mol kg^{-1}$ ,  $DIC = 1586 \mu mol kg^{-1}$ , black symbol in Fig. 3). Figures 3b and c, show how exposed sites, undersaturated in pCO2 and low CH4 concentrations, follow the theoretical trend of photosynthesis and CO<sub>2</sub> release. However, an uptake of CO<sub>2</sub> was both measured and estimated for those sites (Fig. 2), suggesting that gas exchange as release towards the atmosphere could not explain the observed changes in the carbonate system. In contrast, sheltered sites, with supersaturated pCO<sub>2</sub> and elevated CH<sub>4</sub> concentrations, were more scattered around the theoretical respiration line and CO<sub>2</sub> uptake. Gas exchange, as CO<sub>2</sub> uptake, can also be ruled out, as both measured and estimated air-sea fluxes pointed to a net release of CO2 into the atmosphere. Altogether, this suggests that biological processes—primary production at undersaturated pCO<sub>2</sub> sites and respiration at supersaturated sites—mainly influence the carbonate system, while the direct role of air-sea exchange of CO<sub>2</sub> appears to be minor in shaping the inorganic carbonate dynamics across the study area. The similar pattern observed for both pCO2 and CH4 concentrations illustrated the positive relationship between the two gases (Fig. 4), where processes responsible for higher pCO<sub>2</sub>, such as respiration and/or organic matter degradation, also contribute to higher CH<sub>4</sub> concentrations (Reeburgh, 2007). No clear pattern could be observed between the nTA: nDIC ratio and N2O concentration, as expected by the poor Kendall's coefficient of correlation (Fig. 4).

Elevated chl-a concentrations (up to 37.7 μg L<sup>-1</sup>, Fig. 1) in surface seawater were reported across the studied area. Chl-a measurements are typically considered as a proxy for pelagic primary production (Cloern et al., 2014). Therefore, elevated chl-a could indicate active photosynthesis production, resulting in pelagic O2 production and pelagic CO2 uptake. To investigate the relationship between photosynthetic activity and surface water pCO<sub>2</sub>, CH<sub>4</sub>, and N<sub>2</sub>O concentrations, apparent oxygen utilization (AOU), calculated as the difference between oxygen saturation and measured oxygen concentration, was examined with chl-a concentrations. No clear relationship was observed between chl-a and AOU (Fig. 6a). Negative values of AOU, indicative of net photosynthesis, were associated with relatively low chl-a concentrations within the observed range. These negative AOU values corresponded to undersaturated pCO<sub>2</sub> and low CH<sub>4</sub> concentrations (Fig. 6b, c). In contrast, positive AOU values, representing net respiration, were associated with supersaturated pCO2 and elevated CH4 concentrations, both of which increased with higher AOU. A strong correlation exists between AOU and surface water pCO<sub>2</sub> (Fig. 6b), suggesting that pelagic primary production is the main process affecting the carbonate system, as already indicated by the nDIC: nTA ratio (Fig. 3). This relation is weaker for CH<sub>4</sub> and N<sub>2</sub>O, as respiration is not ultimately associated with CH<sub>4</sub> production. Therefore, if chl-a concentration is considered as a proxy for pelagic primary production (Cloern et al., 2014), this suggests that primary production within the pelagic realm is not solely responsible for the change of pCO<sub>2</sub>. The influence of the pelagic realm is likely limited, as phytoplankton biomass is constrained by the shallow water column, and its productivity is low at the end of the summer bloom (Uth et al., 2024). Therefore, processes beyond pelagic production, such as benthic community production and organic matter degradation, are highly likely to influence the observed pCO2 and CH4 dynamics. No clear relationships between AOU and N2O could be observed, as expected from the Kendall's correlation coefficient.

The inputs of freshwater supersaturated in CO<sub>2</sub>, CH<sub>4</sub>, and N<sub>2</sub>O represent a major source of GHGs to the study area. Freshwater mixing with seawater appears to control the concentration of all three GHGs at exposed sites, as much of the variability in surface seawater concentrations can be explained by salinity. In sheltered sites of the archipelago, deviations from the salinitydriven pattern indicate the influence of additional processes. Biological activity is likely the key driver, with pCO2 undersaturated associated with primary production, while pCO2 supersaturated linked to respiration. However, pelagic production alone is unlikely to be the main contributor to the observed changes. Benthic processes such as community production, organic matter degradation, and sediment-water interactions are likely to contribute to both CO2 and CH4 dynamics (Attard et al., 2019). For CH<sub>4</sub>, riverine inputs and subsequent dilution largely determine large-scale patterns, but elevated concentrations within the archipelago reflect localized production and limited oxidation. Large amounts of carbon are turned over in the habitats from the studied area during the summer months (Attard et al., 2019), and macrophyte tissue becomes a direct component of local sediment organic matter pools that favour local CH<sub>4</sub> production (Roth et al., 2022; Wallenius et al., 2021). Anoxic degradation of organic-rich sediments (methanogenesis), exacerbated by warm late-summer waters (Roth et al., 2022; Yvon-Durocher et al., 2014), together with short water residence times that restrict oxidation in both sediment and the overlying water column (Reeburgh, 2007), create favourable conditions for CH<sub>4</sub> accumulation. Similarly, enhanced respiration of organic carbon in shallow sediments elevates CO<sub>2</sub> concentrations (Humborg et al., 2019). While macrophyte growth during summer can draw down pCO2 through photosynthetic CO2 uptake, the subsequent decomposition of deposited tissues may

instead enhance CO<sub>2</sub> and CH<sub>4</sub> release. Spatial variability of N<sub>2</sub>O mirrors the findings of Aalto et al. (2021), with higher concentrations near the Karjaanjoki River linked to allochthonous carbon inputs and elevated nitrate availability. Aalto et al. (2021) suggested that within the archipelago, reduced riverine influence and enhanced sedimentary processes, such as dissimilatory nitrate reduction, likely sustain high ammonium availability and promote active nitrogen recycling between sediments and surface waters, particularly in summer, when autochthonous biomass and sedimentation are highest.

#### 4.2 Air-sea fluxes

Given the strong spatial variability observed in GHG concentrations across the study area, strong variations were also observed in the corresponding air-sea fluxes. In exposed sites between Hanko and TZS, where surface seawater was undersaturated in CO<sub>2</sub> and low in CH<sub>4</sub> concentration, an uptake of atmospheric CO<sub>2</sub> associated with small releases of CH<sub>4</sub> was observed. In contrast, within the archipelago and across Pojoviken Bay, where surface waters were supersaturated in CO2 and exhibited elevated CH<sub>4</sub> concentrations, releases of both CO<sub>2</sub> and CH<sub>4</sub> were observed. Sites 3 and 4 represented exceptions, being the only locations undersaturated in CO<sub>2</sub>, thus acting as a CO<sub>2</sub> sink. Daily CH<sub>4</sub> emissions of ≥ 0.1 mmol m<sup>-2</sup> across all habitats are comparable to, or even higher than, CH<sub>4</sub> fluxes reported from similar (Lundevall-Zara et al., 2021; Roth et al., 2023) or other vegetated coastal ecosystems (Al-Haj and Fulweiler, 2020; Rosentreter et al., 2021). Regarding N<sub>2</sub>O, Pojoviken Bay is estimated to act as a source to the atmosphere. In sheltered sites of the archipelago and exposed sites near TZS, N2O concentrations close to or slightly below atmospheric equilibrium result in small emissions and occasional uptake, respectively. Our estimated air-sea exchanges overall mirrored the spatial trends of chamber-based fluxes and exhibited a similar range of air-sea exchange. Both chamber-based and estimated fluxes may underestimate the air-sea fluxes, as possible ebullition events are not considered. Humborg et al. (2019) suggested that air-sea CH<sub>4</sub> flux is likely dominated by frequent bubbling from the sediment. While this process could be dominant in shallow coastal environments, as shallow seafloor depth promotes a short residence time for CH<sub>4</sub> in the water column, preventing its potential full oxidation as observed in the deep open ocean (Reeburgh, 2007), it was not observed during our chamber-based measurements.

The overall contribution of the study area to atmospheric GHG budgets can be assessed by combining the air-sea flux of CO<sub>2</sub> with the CO<sub>2</sub>-equivalent fluxes of CH<sub>4</sub> and N<sub>2</sub>O. These equivalents are calculated using their respective sustained-flux global warming potential (SGWP), as a greenhouse gas metric to describe the relative radiative impact of a standardized amount of gas over a defined time horizon. Specifically, over a 100-year time horizon, the SGWP of CH<sub>4</sub> and N<sub>2</sub>O is 27 and 273 times, respectively, greater than that of CO<sub>2</sub> (IPCC 2023), on a mass basis, based on:

$$F_{CO_2 - eq_{(X)}} = F_{(X)}SGWP_{(X)} \tag{3}$$

where the  $CO_{2-eq}$  flux of X (CH<sub>4</sub> or  $N_2O$ ),  $F_{CO_2-eq_{(X)}}$ , is the product of the flux of the gas ( $F_{(X)}$ ) and its respective SGWP (i.e., 27 or 273) over the time horizon of 100 years. As direct measurements of air-sea  $N_2O$  fluxes were not available at the time, and to ensure consistency across all three gases, we chose to use the estimated fluxes rather than the chamber-based measurements.

Overall, the study area acted as a net source of GHGs to the atmosphere, with an average release of 1.6 g of CO<sub>2-eq</sub> m<sup>-2</sup> d<sup>-1</sup> over the study period, and ranging from -1.0 g of CO<sub>2-eq</sub> m<sup>-2</sup> d<sup>-1</sup> at S14 to 10.6 g of CO<sub>2-eq</sub> m<sup>-2</sup> d<sup>-1</sup> at the mouth of the Karjaanjoki River (Fig. 7d). Exposed sites between Hanko and TZS generally acted as GHGs sink, primarily due to the substantial CO<sub>2</sub> uptake compensating for CH<sub>4</sub> emissions. An exception was S21, where the small uptake of CO<sub>2</sub> (-0.008 g of CO<sub>2</sub> m<sup>-2</sup> d<sup>-1</sup>) was offset by the release of CH<sub>4</sub> (0.016 g of CO<sub>2-eq</sub> m<sup>-2</sup> d<sup>-1</sup>), resulting in a net source. In Pojoviken Bay, the river mouth and S2 were acting as a strong source of GHGs to the atmosphere, with N<sub>2</sub>O fluxes in the same order of magnitude as those of CH<sub>4</sub>. However, S3 and S4, both sites undersaturated in pCO<sub>2</sub>, remained net GHG sinks, despite releasing similar amounts of CH<sub>4</sub> and N<sub>2</sub>O. In the archipelago, all sheltered sites remained a net source of GHGs to the atmosphere. If, from a climate mitigation perspective, it may seem sufficient to focus on CO<sub>2</sub> (as Figure 7a and d are pretty similar), as in the blue carbon approach, it has been shown that CO<sub>2</sub> uptake can be offset by CH<sub>4</sub> emissions (Roth et al., 2023). Our findings demonstrate that, if CH<sub>4</sub> matters in coastal, nearshore environments, so does N<sub>2</sub>O, since N<sub>2</sub>O fluxes are not only of the same order of magnitude as CH<sub>4</sub> fluxes but can either reinforce the warming effect of CH<sub>4</sub> or partially counterbalance it.

#### 4.3 Conclusions

Shallow coastal ecosystems are highly heterogeneous, with their spatial structure and temporal dynamics of benthic communities shaping ecosystem functions (Snelgrove et al., 2014). This heterogeneity drives strong spatial variability in coastal biogeochemical processes, which exert an important influence on the carbon cycle at both local and global scales (Ward et al., 2020). Yet, despite their importance, identifying the locations and processes regulating coastal CO<sub>2</sub>, CH<sub>4</sub>, and N<sub>2</sub>O fluxes remains uncommon, and global estimates still fail to capture the heterogeneous and dynamic nature of these environments. In particular, northern temperate coastal habitats are poorly represented in current GHG budgets, despite their relative importance: shallow waters (<5 m) in the Baltic Sea cover ~30,000 km² (HELCOM, 2013), an area comparable to 20% of the global distribution of mangroves (Bunting et al., 2018) or seagrass meadows (McKenzie et al., 2020).

Across the study area, freshwater inputs from the Karjaanjoki River and subsequent mixing with seawater largely determined the overall spatial patterns of surface water pCO<sub>2</sub>, CH<sub>4</sub>, and N<sub>2</sub>O concentrations. However, important deviations from this salinity-driven control were observed at the local scale. In sheltered sites within the archipelago, elevated CH<sub>4</sub> concentrations and supersaturated pCO<sub>2</sub> likely reflected benthic processes, including enhanced organic matter respiration and methanogenesis in warm, late-summer shallow waters, where limited oxidation favoured CH<sub>4</sub> accumulation. By contrast, in more exposed sites, mixing processes exerted stronger control, leading to lower GHG concentrations. N<sub>2</sub>O patterns followed riverine influence with higher concentrations near the river mouth associated with inputs of allochthonous carbon and elevated nitrate availability, and lower concentrations within the archipelago reflecting reduced riverine input and active nitrogen recycling between sediments and surface waters (Aalto et al., 2021). Together, these results underline the dual importance of large-scale physical mixing and local benthic processes in shaping the spatial heterogeneity of GHG dynamics. Our findings suggest that benthic ecosystems likely play a pivotal role in regulating GHG dynamics, especially in shallow coastal environments, not only through

https://doi.org/10.5194/egusphere-2025-5068 Preprint. Discussion started: 23 October 2025

© Author(s) 2025. CC BY 4.0 License.

EGUsphere Preprint repository

production but also through consumption pathways, which remain insufficiently constrained in current budgets (Rodil et al., 2021; Roth et al., 2022; Wallenius et al., 2021).

When translated into CO<sub>2-eq</sub>, air—sea GHG fluxes were dominated by CO<sub>2</sub>, while CH<sub>4</sub> and N<sub>2</sub>O contributed comparably but in different ways. CH<sub>4</sub> consistently acted as a source, whereas N<sub>2</sub>O partially offset the CH<sub>4</sub> release through uptake. This interplay emphasizes that the balance between production and consumption processes, particularly within benthic habitats, is critical to understanding coastal contributions to the global carbon budget. Given their extent, northern temperate coastal ecosystems represent a relevant but overlooked source of GHGs, with the potential to amplify the global ocean carbon budget. Our results highlight the urgent need for research that integrates GHG with biodiversity, benthic–pelagic interactions, and microbial processes, and that resolves temporal variability across seasonal cycles. Such knowledge is essential to improve predictions of how coastal ecosystems will mediate carbon–climate feedback under future environmental change.

# Data availability

The complete dataset will be made publicly available upon acceptance of the manuscript.

# **Author contribution**

NXG, AN, and AV designed the hypothesis field sampling. Field measurements were conducted by NXG and AV, laboratory and data analyses were performed by NXG, BD. NXG prepared the manuscript with contributions from all co-authors. All authors approved the final version of the manuscript for submission.

# **Competing interests**

The authors declare no competing interests.

# Acknowledgements

This study was supported by the Jane and Aatos Erkko Foundation (AN, NXG), Walter and Andrée de Nottbeck Foundation (NXG), and the Sophie von Julin's Foundation (AN, AV) and utilized research infrastructure facilities at Tvärminne Zoological Station, University of Helsinki, as part of FINMARI (Finnish Marine Research Infrastructure Consortium). This is a publication from the Centre for Coastal Ecosystem and Climate Change Research (www.coastclim.org). We thank Kurt Spence for his assistance with field work. CH4 and N2O measurements were supported by GreenFeedBack RIA (Greenhouse gas fluxes and earth system feedback) funded by the European Union's HORIZON research and innovation program under grant agreement N°101056921. Views and opinions expressed are, however, those of the authors only and do not necessarily reflect

those of the European Union or CINEA. Neither the European Union nor the granting authority can be held responsible for them. BD is a research associate at the F.R.S-FNRS.

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

Figure 1: Atmospheric conditions: 30-min averaged a) air temperature (°C) and b) wind speed (m/s) recorded at the ICOS station FI-Tvm established at the Tvärminne Zoological Station (TZS). Spatial variation of c) surface seawater temperature (SST, °C), and d) salinity (SSS), e) chlorophyll-a (Chl-a, μg L<sup>-1</sup>) concentration, f) turbidity (FTU), g) colored dissolved organic carbon (CDOM, μg L<sup>-1</sup>), and h) dissolved oxygen saturation (%) across the sampling area. Triangles represent the surface water conditions at the mouth of the Karjaanjoki River.

Figure 2: Left column presents the spatial variation of surface seawater a) pCO<sub>2</sub> (ppm), b) CH<sub>4</sub> (nmol L<sup>-1</sup>), and c) N<sub>2</sub>O concentration (nmol L<sup>-1</sup>). The middle column presents measured air-sea fluxes of d) CO<sub>2</sub> (mmol m<sup>-2</sup> d<sup>-1</sup>) and e) CH<sub>4</sub> (mmol m<sup>-2</sup> d<sup>-1</sup>). The right column presents the calculated air-sea fluxes of f) CO<sub>2</sub> (mmol m<sup>-2</sup> d<sup>-1</sup>), g) CH<sub>4</sub> (mmol m<sup>-2</sup> d<sup>-1</sup>), and h) N<sub>2</sub>O (mmol m<sup>-2</sup> d<sup>-1</sup>). Positive values indicate a release from the sea to the atmosphere, while negative values represent an uptake by the sea. Triangles represent the surface water conditions at the mouth of the Karjaanjoki River.

Figure 3: a) Relationship between salinity, TA (μmol kg<sup>-1</sup>, upward triangles) and DIC (μmol kg<sup>-1</sup>, downward triangles). Relationship between nDIC and nTA, and their association with b) pCO<sub>2</sub> (ppm), c) CH<sub>4</sub> (nmol L<sup>-1</sup>), and d) N<sub>2</sub>O (nmol L<sup>-1</sup>) concentrations. Black symbols represent the seawater endmember. Circles denote exposed sites, and squares denote sheltered sites.

**Figure 4:** Heatmap of Kendall correlation coefficient **a)** between surface water pCO<sub>2</sub>, CH<sub>4</sub>, and N<sub>2</sub>O concentration, and **b)** between main seawater biogeochemical properties (SST, SSS, chl-*a* concentration, colored dissolved organic carbon, turbidity, and O<sub>2</sub> saturation) and pCO<sub>2</sub>, CH<sub>4</sub>, and N<sub>2</sub>O concentration.

a. Heatmap of Kendall correlation between pCO2, CH4, and N2O  $\,$ 

environmental variables and pCO2, CH4, and N2O. SST -0.1905 (p = 0.2422) -0.1238 (p = 0.4550 0.01818 0.8 0.6 -0.4381 (p = 0.0049) SSS -0.81820.4 Chl-a 0.01818(p=1)0.2 0.2095 (p = 0.1966) 0.4095(p = 0.0090) 0.4545 (p = 0.0601) **CDOM** 0.2762(p = 0.0852) -0.2 -0.4 Turb. -0.6 -0.7905 

Figure 5: Relationship between surface seawater salinity and a)  $pCO_2$  (ppm), b)  $CH_4$  (nmol  $L^{-1}$ ), and c)  $N_2O$  (nmol  $L^{-1}$ ) concentrations. In each panel, the dotted line represents the regression across all sites, while the red dashed line shows the regression limited to "exposed" sites only. The blue line in panel a) represents the estimated  $pCO_2$  resulting only from the mixing of the Karjaanjoki River water (S = 0) with seawater (S = 6.42).

Figure 6: Relationship between the apparent oxygen utilization (AOU) and a) surface water Chl-a concentration (μg L<sup>-1</sup>), b) surface water pCO<sub>2</sub> (in ppm), c) CH<sub>4</sub> (nmol L<sup>-1</sup>), and d) N<sub>2</sub>O (nmol L<sup>-1</sup>) associated with the Chl-a concentration (μg L<sup>-1</sup>). The triangle in each panel corresponds to the mouth of the Karjaanjoki River. Circles denote exposed sites, and squares denote sheltered sites.

**Figure 7:** Air-sea fluxes of **a**) CO<sub>2</sub> (g m<sup>-2</sup> d<sup>-1</sup>), and CO<sub>2</sub>-equivalent fluxes of b) CH<sub>4</sub> (g m<sup>-2</sup> d<sup>-1</sup>), and c) N<sub>2</sub>O (g m<sup>-1</sup> d<sup>-1</sup>) estimated using their respective sustained-flux warming potential (SGWPs) over a 100-year time horizon (CH<sub>4</sub> = 27; N<sub>2</sub>O = 273, relative to CO<sub>2</sub> = 1, IPCC 2023), and **d**) the overall budget for all three GHGs (g m<sup>-2</sup> d<sup>-1</sup>).