# Peer review of "Spatial heterogeneity of GHG dynamics across an estuarine ecosystem"

_EGUsphere, 2025_

## Referee Comment (RC2)

Comments on **egusphere-2025-5068**

**Summary**

The manuscript by Geilfus and colleagues presents the results of an extensive campaign devoted to investigate the source/sink dynamics of the major greenhouse gases (GHG) CO2, CH4 and N2O along a temperate estuarine system. The authors aimed to quantify GHG fluxes across the sea-air interface under a salinity gradient, as well as to compare those fluxes between sites with contrasting circulation/atmospheric forcing features. Despite inherent heterogeneity in GHG distribution and sea-air fluxes, the main finding is that while CH4 and N2O mostly offset CO2 uptake in the estuary, N2O might also counteract CH4 emissions.

**General assessment**

Detailed studies of GHG dynamics across the land-ocean continuum are, despite the admittedly increasing attention of the topic over the last decade, still scarce. From that perspective, there is a great value in having such a comprehensive survey of distribution and air-sea fluxes of all three gases simultaneously. An important aspect shown by these measurements/analyses, is that establishing the overall radiative balance of coastal ecosystems is challenging, as departures from the classical view (CO2 uptake coupled to offset from non-CO2 gases) is not necessarily a given, when local-scale environmental variability is considered. While the extent at which, for instance, N2O might offset CH4 emissions in this and other systems, as well as its temporal variability over longer time scales remains to be seen, I do think that it is important to show the complex interactions as it has been done here. Overall, the manuscript is well written, the structure is mostly clear and the figures are of good quality.

That being said, I spotted a few issues which I would invite the authors to address. The perhaps major issue I see, is the difficulty of combining two scales of variability, namely, the spatial gradient due to salinity, and the variability associated to exposed and sheltered areas. While I appreciate that it is challenging to put these different aspects together, I think combining them makes the manuscript less clear than it could be based on the observations. On the one hand, the data presented in this manuscript shows a very clear salinity-dominated gradient which is reflected in other chemical parameters measured along with the gases. On the other, separating areas in exposed and sheltered (which partially overlap with the salinity gradient), adds a complexity that is not necessarily relatable to the fresh-seawater continuum. Another issue I see is that the definitions of exposed and protected are not fully explained in terms of how consistent they are spatially. For instance, while stations 1-4 and 14-21 are clustered together under the exposed category, the GHGs observations do show that the patterns can be radically different (see e.g. Figs. 2 and 7). All in all, my feeling is that splitting the spatial variability in two "chapters" (e.g. i) salinity gradient, ii) sheltered water ways), would help the authors to convey their arguments more clearly.

Another aspect is the discussion on the drivers explaining the observed GHG variability. Due to the nature of the data sets gathered as part of the study, there is an imbalance between the possible depth of interpretation of pelagic vs benthic processes. I appreciate that the authors remain careful in that they suggest benthic sources/sinks are "likely" drivers. However, given the weight of this argument throughout the manuscript, I have to agree with the other reviewer (Truong An Nguyen) in that this part could be improved. Here I wonder whether the authors could make usage of physical information they collected during their survey to compute e.g. a stratification index that could be used to see at what extent the different areas could have been stratified or well mixed. I noticed that mixing was referred to based on literature, but perhaps it would help the interpretation if we have an understanding of water column oxygen and stratification as potential conditions affecting gas

production/consumption/fluxes to the atmosphere. Lastly, I noticed that the authors gave a relatively high weight to AOU as a tracer for explaining GHGs distribution. The problem I see with that is that AOU is not a compelling predictor of e.g. N2O in shallow systems, where air sea gas exchange is dominant. Furthermore, since the connection between primary production (using chlorophyll as a proxy) and N2O is not direct, so I would not expect this to necessarily provide a mechanistic explanation.

In the following I list other comments, some of which substantiate my assessment above, and some of which correspond to specific issues I spotted while reading the manuscript, and that I hope are useful for the authors when preparing their revision.

**Specific comments:**

l. 14 – 21: Given that the crux of the manuscript is illustrating the variability in GHG sources and sinks, it would make sense then to express N2O and CH4 in saturations or Delta values (allowing also the reader to directly draw comparisons with global ocean estimates).

l. 28 – 30: The connection between benthic processes and the statement above (as indicated by "therefore") is not clear.

l. 88 – 100: I would suggest revisiting the categories of exposed and sheltered, in particular because areas grouped within the former seems to behave differently (see also comment above).

l. 102 and ff: Information on calibration and drift of the Li-COR analyzer is missing.

l. 111: Here and in other instance the authors refer to 45 min as the time needed to reach equilibrium. However, from the formulation it is not fully clear whether this is the response time of their equilibrator and how it was quantified. Here a more detailed description of the measurement procedure would be useful for the reader.

l. 130 – 131: The agreement between chamber-derived and calculated flux densities seems to be remarkably good. Since such chambers are known to suffer from artefacts under turbulent conditions, it would be good for the readers (potential users) to learn about the sampling conditions.

l. 155: Here it is not clear whether the standardised wind speeds were computed from the mean of wind speeds at 2 and 3.2 m height.

l. 157 and ff.: I think it would be important, for the sake of clarity, to mention how well the in-situ atmospheric measurements fit/are comparable to the atmospheric measurements at this site. For instance, I noticed that the mean value reported for CO2 (406 ppm; l. 195), is considerably lower than values reported at PAL station at the time of sampling: https://gml.noaa.gov/data/data.php?site=PAL¶meter_name=Carbon%2BDioxide&frequency=Discrete. The values publicly available for PAL oscillate between ca. 409 and 416 ppm for the same time period in 2023. Admittedly, these values are flagged as "preliminary", but even after taking the last year of calibrated (final) measurements and applying the global annual rate of increase as an index (see: https://gml.noaa.gov/ccgg/trends/), the mean atmospheric xCO2 value would be ca. 414 ppm. I did not check for the other gases (and certainly would not expect to change much the overall distribution), but it still would be good to clearly state which values were used and why.

l. 188: This should read "coloured dissolved organic matter".

l. 208 and ff.: Some parts of Fig. 3 are addressed during the results section, while others are mentioned only later during the discussion. Since some of the results are referred to in Fig. 4, it is confusing for the reader. I kindly suggest considering to split Fig. 3 in separate figures that appear then in sections 3 and 4.

l. 210 – 211: I think "air-sea flux densities" would be a more adequate term here (instead of "exchanges").

l. 232 – 241: My impression is that this analysis would be better suited for the discussion. Also, it would perhaps help the line of argumentation if variables which are mechanistically expected to be related are analysed. An example of this would be N2O and CO2.

l. 344 – 365: I think comparing the radiative budget calculated for this system with global estimates (e.g. works by Rosentreter et al and Resplandy et al), would strengthen the arguments laid in this manuscript and emphasize its relevance.

l. 392: It is not clear what is meant by "amplify" the global ocean carbon budget.

Kind regards,
Damian L. Arévalo-Martínez

---

## Author Comment (AC1)

**Reviewer n°3**

Overview:

The manuscript „Spatial heterogeneity of GHG dynamics across an estuarine ecosystem" presents a nice data set of GHG data and related variables in an estuarine ecosystem in the northern Baltic Sea. The data set presents a spatial distribution within the onset of the of the salinity gradient, however, it represents just the late summer period. Nevertheless, the data set included next to the CO2, Methan and N2O as well. The main findings are that the estuarine system is heterogeneous, sinks and sources were found for CO2 and N2O, but methane were supersaturated in all samples and were emitted to the atmosphere.

The manuscript is well written and relatively easy to follow, even if you are not familiar with the study site. Although it is known that the coastal waters and estuaries are important for the GHG balance, comprehensive data set with all three most important GHG are still rare, and they are needed to understand these quite heterogeneous contributions.

Major comments:

If next to CO2 and Methane also N2O is included to understand the interaction between these three GHGs, it would be good to introduce next to the Carbon cycle also the Nitrogen Cycle and the interaction between them. At some points the mentioning of N2O not good connected.

→ We modified the introduction. We can now read starting at L 65:

"$N_2O$ is primarily produced via nitrification (ammonia oxidation) and denitrification (nitrate reduction) in sediments and water columns and is controlled by the availability of dissolved inorganic nitrogen and oxygen (Bange 2006). Eutrophication and hypoxia, often driven by excess nutrient and organic matter inputs, have been shown to promote $N_2O$ emissions (Murray et al., 2015; Brase et al., 2017). Coastal ecosystems are recognized as significant sources of $N_2O$ to the atmosphere (Bange, 2006; Cheung et al., 2025; Resplandy et al., 2024), where denitrification, especially in sediments and on particles, often dominates $N_2O$ production, even in well-oxygenated waters (Wan et al., 2023). However, coastal estimates remain uncertain due to sparse measurements and high spatial heterogeneity (Wan et al., 2023)."

One important variable to understand N2O production and emissions is next to the oxygen supply the availability of nitrate and the other DIN, as the authors mentioned in the discussion. Can you present also at least nitrate concentrations and include them in the correlation matrix?

→ Unfortunately, we did not measure nitrate and other DIN during our sampling. Some measurements were taken in the study area during similar times, but our sampling was not conducted at the same time and/or exact location, making it difficult to use them to improve the interpretation of the $N_2O$ data collected. Maybe our sentence was unclear; we modified it as follows:

L 257:

"Aalto et al. (2021) reported that higher $N_2O$ concentrations were associated with higher nitrate concentrations and inputs of allochthonous carbon, while lower $N_2O$ concentrations were associated with efficient internal recycling of N."

L 355:

"Spatial variability of $N_2O$ mirrors the findings of Aalto et al. (2021), who linked higher $N_2O$ concentrations near the Karjaanjoki River to higher nitrate inputs and allochthonous carbon inputs."

Somewhere in the manuscript the authors must explain what the potential sinks for N2O are and present an explanation for the undersaturation of the N2O. Do you assume that N2O is consumed under anaerobic conditions in the sediments? Are the also processes in the water column which consumed N2O.

→ The lack of nitrate measurements associated with our observations makes it difficult to explain the observed trend, but a study by Aalto et al. (2021) from the same area, along the same salinity gradient, suggested that sites located in the 'offshore archipelago,' where autochthonous (locally produced, e.g., phytoplankton-derived) organic matter is more common and nitrate concentrations are lower, promote DNRA as the main pathway for recycling nitrate to ammonium without producing $N_2O$. This process acts as a sink by suppressing $N_2O$ formation. They also suggest that complete denitrification occurred in environments with a high organic carbon-to-nitrate ratio, such as those with abundant autochthonous organic matter.

We modified our manuscript and can now read from L 353:

"Spatial variability of $N_2O$ mirrors the findings of Aalto et al. (2021), who linked higher $N_2O$ concentrations near the Karjaanjoki River to higher nitrate inputs and allochthonous carbon inputs. They suggested that the ratio between nitrate and autochthonous organic carbon controls the balance between N-removing denitrification and N-recycling through Dissimilatory Nitrate Reduction to Ammonium (DNRA), as well as the end-product of denitrification (Aalto et al., 2021). Within the archipelago, where riverine influence is limited, DNRA can produce significant amounts of bioavailable ammonium, enhancing nitrogen recycling between sediments and surface water, especially in summer, when autochthonous biomass production and sedimentation are highest."

The aim of the study which is formulated at the end of the introduction seems to be a bit too ambiguous. I would suggest orientating the aims a bit more on the structure of the discussion section.

→ We rephrased the last chapter of the introduction (L 77). It now reads:

"We investigate the dynamics of $CO_2$, $CH_4$, and $N_2O$ (GHGs) along a salinity gradient and across contrasting coastal habitats within an estuary. We combined detailed field measurements of surface seawater physical and biogeochemical properties with both in situ measurements and calculated estimates of air–sea GHG exchange. This approach allowed us to capture spatial variability and discuss the roles of physical drivers and biological processes in the observed changes in GHG concentrations. In doing so, we aim to estimate the contribution of coastal ecosystems to GHG emissions and provide additional data that may be useful in improving global estimates of GHG emissions from the coastal ocean."

The explanation of the importance of the sediments for the GHG production, without having direct measurements comes a bit out of the blue and contradicts the hypotheses that the GHGs are mainly just mixed from the river inputs.

→ We have tuned down the importance of the sediments for GHG production, as we don't have direct measurements. In the new section 4.3 Exposed and semi-sheltered vs sheltered sites, we can read at L 338:

"Freshwater mixing with seawater appears to control the concentration of all three GHGs at exposed and semi-sheltered sites, as much of the variability in surface seawater concentrations can be explained by salinity. In sheltered sites of the archipelago, deviations from the salinity-driven pattern indicate the influence of additional processes."

Further down, L 349, we can read:

"Anoxic degradation of organic-rich sediments (methanogenesis), exacerbated by warm late-summer waters (Roth et al., 2022; Yvon-Durocher et al., 2014), combined with short water residence times that limit oxidation in both sediment and the overlying water column (Reeburgh, 2007), creates favourable conditions for $CH_4$ production. Due to the sheltered nature of the sites and limited water exchange, the produced $CH_4$ can accumulate. Similarly, enhanced respiration of organic carbon in shallow ecosystems elevates $CO_2$ concentrations (Humborg et al., 2019)."

Some specific / minor comments:

Abstract: quite too long and can be more focused.

→ We tried to shorten it, taking into consideration comments from other reviewers.

L21: N2O is missing

→ We are not sure about what the reviewer is referring to.
L 21 reads: "$CH_4$ concentrations were consistently supersaturated (19 to 469 nmol $L^{-1}$) compared to the atmosphere, resulting in a net source to the atmosphere from 0.014 to 1.39 mmol $m^{-2}$ $d^{-1}$."
This sentence is not related to $N_2O$, and the previous sentence clearly refers to both $CO_2$ and $N_2O$.

L 28: Is Methane always a source and N2O a sink?

→ The sentence has been changed as follows: "The overall budget of air–sea GHG exchanges was dominated by $CO_2$ fluxes, with $CH_4$ consistently acting as a source, and $N_2O$ alternating between source and sink."

L64. There is some literature that N2O production and Oxygen depletion correlated, e.g. in the Elbe Estuary

→ We modified this section; see earlier response, where we mentioned studies on $N_2O$ dynamics in the Elbe Estuary.

L 88: What are soft-sediment habitats

→ The sentence now reads (L 93):

"Sampling sites (N = 21) were selected to encompass a wide range of soft-sediment habitats (e.g., including both vegetated and non-vegetated sediments with grain size ranging from coarse sand to clay, silt, and mud) and to represent a spatial gradient (50 km) from the outer to innermost archipelago."

L100. What was the partial sampling strategy? From fresh to salty? Each Stations just once, how much on one day….

→ We added the following precision (L 110):

"Sites were visited once during the study period, with sampling conducted every two to three days, depending on weather conditions and boat availability."

L231: Did you measure DIN (Nitrate, …)

→ Unfortunately, we did not measure nitrate and other DIN during our sampling. Some measurements were taken in the study area during similar times, but our sampling was not conducted at the same time and/or exact location, making it difficult to use them to improve the interpretation of the $N_2O$ data collected.

L 306 ff. Here comes the main concrete results of the manuscript that can be one new chapter

→ We have not split section "4. Discussion" into four subsections:
  4.1. Salinity gradient
  4.2 Biological driver
  4.3 Exposed and semi-sheltered *vs* sheltered sites (the new chapter as requested)
  4.4 Air-sea flux densities

---

## Author Comment (AC2)

**Reviewer n°1:**

**Overview**

This manuscript (egusphere-2025-5068) presents a valuable dataset on greenhouse gas (GHG) dynamics across 21 sites in a temperate estuary. The study's primary strength lies in providing concurrent measurements of all three major GHGs across a spatial gradient from the river mouth to the outer archipelago. The spatial coverage, which captures contrasting habitats (sheltered vs. exposed sites), provides a useful map of GHG hotspots and sinks. However, the manuscript has several methodological flaws and some overinterpretation of results. The authors claim that benthic processes drive observed patterns, despite the lack of direct benthic measurements.

The authors have assembled impressive fieldwork and an extensive measurement campaign. Reframing the manuscript to emphasize its strengths (comprehensive spatial coverage, simultaneous GHG measurements) while providing more details for the method section and acknowledging some limitations (no benthic measurements, discrete sampling) will strengthen the work considerably.

Thank you for your constructive comments. We have addressed them below.

**Major Concerns**

1. The manuscript repeatedly mentions "benthic processes" and "methanogenesis", but there are no direct benthic measurements. The authors' attempt to deconvolve contributions using Apparent Oxygen Utilization (AOU; Figure 6) is insufficient because AOU reflects the net result of all processes (pelagic, benthic, and advective) and cannot isolate the benthic contribution. The authors observe deviations in surface water GHG concentrations from expected salinity-driven patterns and infer benthic processes, but are not really convinced.

→ This is correct; we don't have direct measurements of benthic processes and/or methanogenesis. To avoid misunderstanding, we have now modified both the abstract and the introduction to place less emphasis on these unmeasured processes. We do, however, still use Apparent Oxygen Utilization (AOU) to discuss the potential importance of benthic processes relative to pelagic contributors. We recognize that AOU may not be perfect because it reflects all processes that affect $O_2$ concentration in the water column. Nonetheless, we believe AOU can provide valuable insights, given the poor correlation ($R^2 = 0.03$) between chl-a and AOU. Using chl-a as a proxy for pelagic primary production, this suggests that primary producers from the pelagic realm have a limited impact on the AOU. If pelagic processes alone cannot explain the observed changes in GHGs, we suggest that processes beyond the pelagic realm, such as benthic community production and organic-matter degradation, are likely contributing to the observed surface-water $pCO_2$, $CH_4$, and $N_2O$ concentrations. This is further supported by studies from Attard et al. (2019) and Roth et al. (2023) (performed at similar water depth and in the same area), who highlight how benthic processes such as community production, organic matter degradation, and sediment-water interactions are likely to influence both $CO_2$ and $CH_4$ dynamics.

The discussion has been adjusted to better reflect our approach while maintaining a more tempered tone in both the discussion and the conclusion.

2. Unclear gas transfer velocity and fluxes calculation. The authors measured $CO_2$ and $CH_4$ fluxes directly using floating accumulation chambers. However, for the final $CO_2$-equivalent budget in Figure 7, they state they "chose to use the estimated fluxes" (line 350-354). The authors adopted a gas transfer velocity (k) wind parameterization, despite having direct measured flux data, water-phase partial pressure and partial pressure data. These three data points are what are needed to derive their own site-specific gas transfer velocity. With these measurements, the authors would have generated a novel, site-specific parameterization, a valuable contribution for gas transfer velocity parameterizations. The authors' justification that Randers Fjord is most comparable (line 148) is insufficient. Additionally, several paragraphs in the discussion are more closely related to the method section.

→ Thank you for this comment. You are absolutely right that, based on our direct measurements of surface water $pCO_2$ and $pCH_4$ and the corresponding air–sea exchange, we could theoretically derive a site-specific gas transfer coefficient (k). However, it is important to note that we conducted only a small number of chamber measurements at each site. The reported values, therefore, represent average air–sea fluxes. We have clarified this in the manuscript, which now states at line 149:

"Reported fluxes correspond to the mean of two to three individual measurements conducted at each site."

As noted in Borges et al. (2004a) and mentioned in the manuscript, the formulation of k is often site-specific. Consequently, deriving a k within the scope of this study would require estimating one for each site we visited. However, we currently lack direct site-specific wind measurements, which are essential for deriving a robust k. The limited number of air–sea exchange measurements we collected is hence insufficient to support a

reliable estimation. Finally, water turbulence is known to be a key parameter controlling k (Borges et al., 2004a; 2004b), and this variable was not measured. For all these reasons, we believe that estimating k under these conditions would not be meaningful.

We are aware of several estimates of k in estuarine environments: i) Borges et al. (2004a) for Randers Fjord, the Scheldt, and the Thames; ii) Borges et al. (2004b) for the Scheldt; and iii) Jiang et al. (2008) for the Georgia estuaries. As previously mentioned, k is highly dependent on water turbulence (Borges et al., 2004a; 2004b), which can be amplified by tidal currents. Among these studies, only Randers Fjord experiences minimal tidal influence. One specific aspect of working in the Baltic Sea is the absence of tides. Therefore, since we do not experience tides at our sites, we used the k value established for Randers Fjord. This reasoning is mentioned in the manuscript on lines 156–166.

Borges Alberto Vieira Delille Bruno Schiettecatte Laure-Sophie Gazeau Frédéric Abril Gwenaöl Frankignoulle Michel , (2004a), Gas transfer velocities of CO2 in three European estuaries (Randers Fjord, Scheldt, and Thames), Limnology and Oceanography, 5, doi: 10.4319/lo.2004.49.5.1630.

Borges, A.V., Vanderborght, JP., Schiettecatte, LS. et al. Variability of the gas transfer velocity of CO2 in a macrotidal estuary (the Scheldt). Estuaries 27, 593–603 (2004b). https://doi.org/10.1007/BF02907647

Jiang Li-Qing Cai Wei-Jun Wang Yongchen , (2008), A comparative study of carbon dioxide degassing in river- and marine-dominated estuaries, Limnology and Oceanography, 53, doi: 10.4319/lo.2008.53.6.2603.

3. The authors describe "in situ" measurement using a custom flow-through system (line 104) but then detail a protocol where they stopped at 21 discrete sites, waiting "until equilibrium was reached (up to 45 minutes)". This equilibration time is quite long for the LI-7810 analyzer, which has a response time of 2 seconds. Please provide more details on the "custom-built flow-through system" and why it is turning a high-resolution instrument into a 21-point discrete sampler.

→ Indeed, we used a custom-built flow-through system where water is pumped through multiple sensors. At each site, water was continuously circulated, and all parameters were measured. While most parameters could be recorded quickly, gas measurements, especially for $CH_4$, took noticeably more time. Our setup relies on continuous measurement of the gas phase, which equilibrates with water through an equilibrator. Therefore, the speed of measurements depends, among other factors (such as the size and type of the equilibrator), on the gas solubility. While $CO_2$ equilibrium was reached within minutes, $CH_4$ solubility caused equilibration to take much longer, particularly at high concentrations, such as those observed at Site 1, where $xCH_4$ values reached up to 282 ppm.

This issue occurs not only when moving from low to high concentrations but also when shifting from high to low concentrations. Since our study examines the spatial variability of GHG concentrations, it was important to ensure that each measurement truly reflected an equilibrium state. Therefore, to ensure high-quality data, we allowed enough time—up to 45 minutes—for the system to reach complete equilibrium with the water.

Finally, the 2-second response time mentioned by the reviewer is correct and appears in the instrument manual (https://www.licor.com/support/LI-7810/topics/specifications.html#top). However, this response time applies only once the gas is already in the detector and only within the range of 0–2 ppm $CH_4$. In our case, we first had to extract the gas from the water using the equilibrator, a process controlled by gas solubility, and our concentrations were well above the range specified in the manual.

We added this precision in the manuscript at line 120:

"Equilibration between the seawater and gas phases was monitored in real time using a laptop connected to the IRGA. Equilibrium was considered reached when both $CO_2$ and $CH_4$ concentrations stabilized at a clear plateau. Depending on the concentration gradient between sites, this equilibration period could vary substantially. While $CO_2$ typically reached equilibrium within a few minutes, $CH_4$ required longer times (up to 45 minutes) to reach a stable plateau."

**Specific Comments**

**Line 8:** "Estuaries remain understudied for GHGs" is overstated. Please moderate language to acknowledge the growing body of estuarine GHG research. Recent literature demonstrates substantial research on estuarine GHG emissions, as shown below.

→ We changed the sentence, taking also into consideration the input from reviewer 2 mentioning that "Detailed studies of GHG dynamics across the land-ocean continuum are, despite the admittedly increasing attention of the topic over the last decade, still scarce. From that perspective, there is a great value in having such a comprehensive survey of distribution and air-sea fluxes of all three gases simultaneously."

Line 8 now reads:

"Although they have gained more attention in the past decade, detailed studies of GHG dynamics across the land-ocean continuum, including shallow-water estuaries, remain relatively scarce even though they are active sources of the three most potent greenhouse gases (GHGs): carbon dioxide ($CO_2$), methane ($CH_4$), and nitrous oxide ($N_2O$)."

**Line 29:** "$CH_4$ and $N_2O$ contributed differently as a source and a sink". This phrasing is wrong. Data show $CH_4$ was consistently supersaturated. Only $CO_2$ and $N_2O$ acted as both source and sink.

→ We changed the sentence as follows (L29):

"The overall budget of air–sea GHG exchanges was dominated by $CO_2$ fluxes, with $CH_4$ consistently acting as a source, and $N_2O$ alternating between source and sink."

**Line 115:** Quite a complete setup, but not sure why there is a complete absence of pH measurements, which significantly limits the validation of carbonate chemistry calculations.

→ Indeed, it would have been valuable to include pH measurements. Unfortunately, we did not have the capability to measure pH at the time. However, we collected discrete water samples for TA and DIC, which allow us to better characterize and understand the carbonate system dynamics during the sampling period.

**Lines 125-130:** Were these floating chambers anchored or drifting (following the current)?

→ The manuscript reads:

"Air-sea exchanges of $CO_2$ and $CH_4$ were measured using the accumulation chamber technique (Frankignoulle, 1988). The chamber consists of a polyethylene container (internal diameter: 34 cm, height: 14 cm, total volume = 11.4 L) connected in a closed loop to the IRGA."

Because the chambers needed to stay connected to the IRGA, the chamber was kept 'anchored' to the boat. We used 1.5 m of Tygon tubing to connect the chamber to the IRGA, allowing the chamber to float freely on the water surface while remaining close enough to ensure safe measurements for the IRGA (e.g., preventing water from entering the IRGA), especially when other boats passed nearby. The air volume of the tubing was included in the flux calculations.

We modified the text as (L140):

"Air-sea exchanges of $CO_2$ and $CH_4$ were measured using the accumulation chamber technique (Frankignoulle, 1988). The chamber consists of a polyethylene container (internal diameter: 34 cm, height: 14 cm, total volume = 11.4 L) connected in a closed loop to the IRGA with 1.5 m long Tygon tubing, which allows the chamber to move freely on the water surface."

**Line 134:** The choice to use Borges et al. (2004) parameterization instead of deriving site-specific values is unjustified. I suggest comparing the K600 derived from the floating chamber with the K600 by Borges et al. (2004).

→ Please refer to our response provided earlier on our decision to use the k estimated by Borges et al (2004) for the Randers fjord.

**Line 160:** If the system was stopped for 45 minutes at each station (line 118), why would there be 'sharp concentration changes' or 'data from transition periods between stations'? This statement is really confused about the actual sampling protocol.

→ We modified L 160 as follows:

"The recorded data from the IRGA for both seawater and air-sea flux measurements were filtered to remove measurements taken during the transition between stations and when switching between ambient air and seawater measurements, as the IRGA requires time to respond to sharp concentration changes. Additionally, data affected by improper functioning (i.e., seawater flow < 1.5 L min–1) were discarded."

This implies that the IRGA was used for both seawater and air–sea flux measurements. Consequently, when switching the system from the seawater flow-through setup to the air–sea flux measurements, in which atmospheric concentrations were recorded, large changes in gas concentrations were observed. Please refer to our earlier response describing our system.

**Lines 210-2015:** The Results section for GHGs is very thin on statistical description. The Kendall correlation analysis is currently in the Discussion, but is a presentation of results. It should be moved to the Results section.

→ We have moved the statistical description from the Discussion to the Results section.

**Figure 5:** The "seawater endmember" has a salinity of only 6.36. This is very low for a seawater endmember. What is the salinity range in that region?

→ Salinity in the area ranged from approximately 0 (near the river inputs) to about 6.5 toward Hanko, as described in the manuscript (Fig. 1). This salinity range is expected for the area and is consistent with the values reported in Lehmann et al. (2022). Below is a visual representation of the sea surface salinity (1.5 m depth), illustrating the pronounced salinity gradient across the Baltic Sea. This map is based on hydrodynamic model data from Lehmann et al. (2022).

[Figure]

The map is from Jaspers, C.; Bezio, N.; Hinrichsen, H.-H. Diversity and Physiological Tolerance of Native and Invasive Jellyfish/Ctenophores along the Extreme Salinity Gradient of the Baltic Sea. Diversity 2021, 13, 57. https://doi.org/10.3390/d13020057 (https://www.mdpi.com/1424-2818/13/2/57). The red square represents our research area.

Lehmann, A., Myrberg, K., Post, P., Chubarenko, I., Dailidiene, I., Hinrichsen, H.-H., Hüssy, K., Liblik, T., Meier, H. E. M., Lips, U., and Bukanova, T.: Salinity dynamics of the Baltic Sea, Earth Syst. Dynam., 13, 373–392, https://doi.org/10.5194/esd-13-373-2022, 2022.

**Lines 249-256 (Mixing Model) and Line 268 (nTA/nDIC):** Details about theoretical $pCO_2$ calculations from conservative mixing and alkalinity/DIC normalization procedures belong in Methods, not Discussion.

→ The lines related to the mixing model read:

"Assuming that changes in SSS reflect a mixing ratio between the Karjaanjoki River and the seawater endmembers and based on the observed linear relationships between salinity and both TA and DIC (Fig. 3a), representing conservative mixing between the two water masses, TA and DIC can be estimated based on salinity. The estimated TA and DIC are then used to compute $pCO_2$ (blue dashed line, Fig. 5a) using CO2SYS v3 (Sharp et al., 2023), applying the carbonic acid dissociation constants (K1 and K2) of Millero et al. (2006) and the KHSO4 formulation of Dickson (1990)."

Lines related to nTA/nDIC read:

"Therefore, to discard the impact of freshwater inputs, TA and DIC were normalized to the average surface seawater salinity of 5 (denoted as nTA and nDIC, respectively) using the normalization of Friis et al. (2003)."

We believe that moving this information to the Methods section would be inappropriate because it would require the reader to infer why these steps are included. Keeping this information in its current location does not disrupt the manuscript's flow and remains sufficiently concise.

**Some GHG research papers for estuarine systems:**

Yeo, J. Z. Q., Rosentreter, J. A., Oakes, J. M., Schulz, K. G., & Eyre, B. D. (2024). High carbon dioxide emissions from Australian estuaries driven by geomorphology and climate. Nature communications, 15(1), 3967.

Zheng, Y., Wu, S., Xiao, S., Yu, K., Fang, X., Xia, L., ... & Zou, J. (2022). Global methane and nitrous oxide emissions from inland waters and estuaries. *Global Change Biology*, *28*(15), 4713-4725.

Nguyen, A. T., Némery, J., Gratiot, N., Dao, T. S., Le, T. T. M., Baduel, C., & Garnier, J. (2022). Does eutrophication enhance greenhouse gas emissions in urbanized tropical estuaries?. *Environmental Pollution*, *303*, 119105.

Borges, A. V., Abril, G., & Bouillon, S. (2018). Carbon dynamics and $CO_2$ and $CH_4$ outgassing in the Mekong Delta. *Biogeosciences*, *15*(4), 1093-1114.

**Reviewer n°2:**

**Summary**

The manuscript by Geilfus and colleagues presents the results of an extensive campaign devoted to investigate the source/sink dynamics of the major greenhouse gases (GHG) $CO_2$, $CH_4$, and $N_2O$ along a temperate estuarine system. The authors aimed to quantify GHG fluxes across the sea-air interface under a salinity gradient, as well as to compare those fluxes between sites with contrasting circulation/atmospheric forcing features. Despite inherent heterogeneity in GHG distribution and sea-air fluxes, the main finding is that while $CH_4$ and $N_2O$ mostly offset $CO_2$ uptake in the estuary, $N_2O$ might also counteract $CH_4$ emissions.

**General assessment**

Detailed studies of GHG dynamics across the land-ocean continuum are, despite the admittedly increasing attention of the topic over the last decade, still scarce. From that perspective, there is a great value in having such a comprehensive survey of distribution and air-sea fluxes of all three gases simultaneously. An important aspect shown by these measurements/analyses, is that establishing the overall radiative balance of coastal ecosystems is challenging, as departures from the classical view ($CO_2$ uptake coupled to offset from non-$CO_2$ gases) is not necessarily a given, when local-scale environmental variability is considered. While the extent at which, for instance, $N_2O$ might offset $CH_4$ emissions in this and other systems, as well as its temporal variability over longer time scales remains to be seen, I do think that it is important to show the complex interactions as it has been done here.

→ Thank you.

Overall, the manuscript is well written, the structure is mostly clear and the figures are of good quality. That being said, I spotted a few issues which I would invite the authors to address. The perhaps major issue I see, is the difficulty of combining two scales of variability, namely, the spatial gradient due to salinity, and the variability associated to exposed and sheltered areas. While I appreciate that it is challenging to put these different aspects together, I think combining them makes the manuscript less clear than it could be based on the observations. On the one hand, the data presented in this manuscript shows a very clear salinity-dominated gradient which is reflected in other chemical parameters measured along with the gases. On the other, separating areas in exposed and sheltered (which partially overlap with the salinity gradient), adds a complexity that is not necessarily relatable to the fresh-seawater continuum. Another issue I see is that the definitions of exposed and protected are not fully explained in terms of how consistent they are spatially. For instance, while stations 1-4 and 14-21 are clustered together under the exposed category, the GHGs observations do show that the patterns can be radically different (see e.g. Figs. 2 and 7).

→ We re-evaluated our definitions of exposed and sheltered sites. Lammerant et al (2025) studied the same sites and categorized the sampling locations into four exposure-salinity groups: outer (exposed), middle (semi-sheltered), inner area (sheltered), and Pojo Bay (see their fig. 1).

The following table displays our original classification, Lammerant's, and the new classification used in our manuscript.

| Site number | Initial classification | Lammerant et al (2025) | New classification |
|---|---|---|---|
| 1 | Exposed | Pojo Bay | Exposed |
| 2 | Exposed | Pojo Bay | Exposed |
| 3 | Exposed | Pojo Bay | Exposed |
| 4 | Exposed | Pojo Bay | Exposed |
| 5 | Sheltered | Pojo Bay | Sheltered |
| 6 | Sheltered | Pojo Bay | Semi-sheltered |
| 7 | Sheltered | Sheltered | Sheltered |
| 8 | Sheltered | Sheltered | Sheltered |
| 9 | Sheltered | Sheltered | Sheltered |
| 10 | Sheltered | Sheltered | Sheltered |
| 11 | Sheltered | Sheltered | Sheltered |
| 12 | Exposed | Semi-sheltered | Semi-sheltered |
| 13 | Sheltered | Semi-sheltered | Semi-sheltered |
| 14 | Exposed | Semi-sheltered | Semi-sheltered |
| 15 | Exposed | Semi-sheltered | Semi-sheltered |
| 16 | Exposed | Semi-sheltered | Semi-sheltered |
| 17 | Exposed | Exposed | Exposed |
| 18 | Exposed | Exposed | Exposed |

| 19 | Exposed | Exposed | Exposed |
|----|---------|---------|---------|
| 20 | Exposed | Exposed | Exposed |
| 21 | Exposed | Exposed | Exposed |

To align with Lammerant's study, we adapted our manuscript to Lammerant's classification, except for Pojo Bay, which is a geographic location and not indicative of exposure level. Sites within Pojo Bay (sites 1 to 4) are broadly open to the bay and are directly influenced by wind and water input from both the Karjaajoki River and the archipelago. Consequently, those sites will remain labeled as 'exposed'. Sites 5 and 6 are not actually situated in Pojo Bay.

Site 5 is situated within a small bay and connected to the main water channel through a relatively narrow opening. As a result, it will remain labeled as 'sheltered'. Site 6 is somewhat protected from the main channel with features similar to those of sites labeled 'semi-sheltered' in Lammerant's study. Consequently, we decided to change its label from 'sheltered' to 'semi-sheltered'.

We modified the text, and now L 93 reads:

"Sampling sites (N = 21) were selected to encompass a wide range of soft-sediment habitats (e.g., including both vegetated and non-vegetated sediments with grain size ranging from coarse sand to clay, silt, and mud) and to represent a spatial gradient (50 km) from the outer to innermost archipelago. We used the classification by Lammerant et al. (2025), who categorize sampling sites based on their salinity and sediment characteristics to indicate the degree of exposure to environmental forces. Exposed sites (sites 1–4 and 17–21) are likely to experience stronger wave, wind, and current energy, as well as higher water exchange, compared to sheltered sites. Sites within Pojo Bay (sites 1-4) are broadly open to the bay and are directly influenced by wind and water input from the Karjaajoki River and the archipelago. Therefore, those are included with the exposed sites. The sheltered sites (sites 5 and 7–11) are more enclosed and likely to have limited water circulation. Semi-sheltered sites fall between exposed and sheltered environments (sites 6 and 12–16)."

From a GHG perspective, even though sites 1–4 and 17–21 are labeled as "exposed'' due to the clear salinity gradient, their GHG patterns can still vary substantially, as the reviewer noted. We hope that our revised explanation of the classification—from exposed to sheltered—and how it relates to water exchange and exposure, now makes our approach clearer.

Figures 3, 5, and 6 have been modified to reflect this change of categories.

All in all, my feeling is that splitting the spatial variability in two "chapters" (e.g. i) salinity gradient, ii) sheltered water ways), would help the authors to convey their arguments more clearly.

→ Thank you for the suggestion. We divided the discussion into different sections, and added a new one based on the recommendation from reviewer n°3.

4.1 Salinity gradient

4.2 Biological drivers

4.3 Exposed and semi-sheltered *vs* sheltered sites

4.4 Air-sea flux densities

Another aspect is the discussion on the drivers explaining the observed GHG variability. Due to the nature of the data sets gathered as part of the study, there is an imbalance between the possible depth of interpretation of pelagic vs benthic processes. I appreciate that the authors remain careful in that they suggest benthic sources/sinks are "likely" drivers. However, given the weight of this argument throughout the manuscript, I have to agree with the other reviewer (Truong An Nguyen) in that this part could be improved.

→ Here is what we answered to reviewer 1 (Truong An Nguyen):

This is correct; we don't have direct measurements of benthic processes and/or methanogenesis. To avoid misunderstanding, we have now modified both the abstract and the introduction to place less emphasis on these unmeasured processes. We do, however, still use Apparent Oxygen Utilization (AOU) to discuss the potential importance of benthic processes relative to pelagic contributors. We recognize that AOU may not be perfect because it reflects all processes that affect $O_2$ concentration in the water column. Nonetheless, we believe AOU can provide valuable insights, given the poor correlation ($R^2 = 0.03$) between chl-a and AOU. Using chl-a as a proxy for pelagic primary production, this suggests that primary producers from the pelagic realm have a limited impact on the AOU. If pelagic processes alone cannot explain the observed changes in GHGs, we suggest that processes beyond the pelagic realm, such as benthic community production and

organic-matter degradation, are likely contributing to the observed surface-water $pCO_2$, $CH_4$, and $N_2O$ concentrations. This is further supported by studies from Attard et al. (2019) and Roth et al. (2023) (performed at similar water depth and in the same area), who highlight how benthic processes such as community production, organic matter degradation, and sediment-water interactions are likely to influence both $CO_2$ and $CH_4$ dynamics.

The discussion has been adjusted to better reflect our approach while maintaining a more measured tone in both the discussion and the conclusion.

Here I wonder whether the authors could make usage of physical information they collected during their survey to compute e.g. a stratification index that could be used to see at what extent the different areas could have been stratified or well mixed. I noticed that mixing was referred to based on literature, but perhaps it would help the interpretation if we have an understanding of water column oxygen and stratification as potential conditions affecting gas production/consumption/fluxes to the atmosphere.

→ Thank you for the inputs. Unfortunately, we measured only surface seawater properties, which makes it difficult to investigate stratification or mixing processes at the time of sampling.

Lastly, I noticed that the authors gave a relatively high weight to AOU as a tracer for explaining GHGs distribution. The problem I see with that is that AOU is not a compelling predictor of e.g. $N_2O$ in shallow systems, where air sea gas exchange is dominant. Furthermore, since the connection between primary production (using chlorophyll as a proxy) and $N_2O$ is not direct, so I would not expect this to necessarily provide a mechanistic explanation.

→ AOU is mainly used to discuss the role of biological processes and highlight the potential importance of benthic over pelagic contributors. We recognize that AOU may not be ideal, but it offers an interesting perspective (see earlier answer above).

The relationship between AOU and $N_2O$, initially reported in the manuscript, was incorrect due to an issue in our code, which caused the figure to be incorrect. We have rectified it, and it seems that AOU is quite well correlated with $N_2O$ ($R^2 > 0.7$).

We modified Figure 6, and we can now read at L 330:

"Despite the lack of correlation between $N_2O$ and chl-a or $O_2$ concentrations (Fig. 4), $N_2O$ exhibited a clear positive correlation with AOU ($R^2 > 0.7$). This correlation is known to reflect the coupling between $O_2$ consumption (from organic matter remineralization) and microbial nitrogen cycling processes that produce $N_2O$ (Kock et al., 2016; Carrasco et al., 2017)."

In the following I list other comments, some of which substantiate my assessment above, and some of which correspond to specific issues I spotted while reading the manuscript, and that I hope are useful for the authors when preparing their revision.

**Specific comments:**

**l. 14 – 21:** Given that the crux of the manuscript is illustrating the variability in GHG sources and sinks, it would make sense then to express $N_2O$ and $CH_4$ in saturations or Delta values (allowing also the reader to directly draw comparisons with global ocean estimates).

→ Those lines read: "Surface water $pCO_2$ and $N_2O$ concentration ranged from undersaturated (160 ppm and 9 nmol $L^{-1}$, respectively) to supersaturated (2521 ppm and 25 nmol $L^{-1}$, respectively), compared to the atmosphere, resulting in an uptake of -36 and -0.0021 mmol $m^{-2}$ $d^{-1}$, and a release up to 220 and 0.0383 mmol $m^{-2}$ $d^{-1}$, respectively. $CH_4$ concentrations were consistently supersaturated (19 to 469 nmol $L^{-1}$) compared to the atmosphere, resulting in a net source to the atmosphere from 0.014 to 1.39 mmol $m^{-2}$ $d^{-1}$."

These lines effectively show the observed concentrations of the three gases, highlight their under- and supersaturation relative to the atmosphere, and are enough for the abstract section.

However, we modified the Results section, where we included the complete range of supersaturation for $CH_4$, L 216 now reads:

"Surface water $CH_4$ concentrations were consistently supersaturated (from 636 to 14609 %, data not shown)…"

For $N_2O$, L 222 now reads:

"$N_2O$ concentration ranged from 9 to 25 nmol $L^{-1}$, with saturation level ranging from 93 to 255 %. Higher concentrations and maximum saturation were observed at the mouth of the Karjaanjoki River (Fig. 2c)."

**l. 28 – 30:** The connection between benthic processes and the statement above (as indicated by "therefore") is not clear.

→ We deleted the word "therefore".

**l. 88 – 100**: I would suggest revisiting the categories of exposed and sheltered, in particular because areas grouped within the former seem to behave differently (see also comment above).

→ See our previous answer above. We revised the categories of exposed and sheltered sites based on published work from Lammerant et al (2025).

**l. 102 and ff:** Information on calibration and drift of the Li-COR analyzer is missing.

→ You are correct. The LI-COR was factory calibrated before the fieldwork, and standard gases for $CO_2$ (150, 420, and 1500 ppm) and $CH_4$ (1, 20, and 150 ppm) were passed through the IRGA both before and after deployment.

This precision has been added to the manuscript L 126, now reads: "The IRGA was factory-calibrated, and standard gases for $CO_2$ (150, 420, and 1500 ppm) and $CH_4$ (1, 20, and 150 ppm) were passed through the IRGA both before and after deployment. The data were correct for potential drift."

**l. 111:** Here and in other instances the authors refer to 45 min as the time needed to reach equilibrium. However, from the formulation, it is not fully clear whether this is the response time of their equilibrator and how it was quantified. Here, a more detailed description of the measurement procedure would be useful for the reader.

→ While most parameters could be recorded quickly, gas measurements, especially for $CH_4$, took noticeably more time. Our setup relies on continuous measurement of the gas phase, which equilibrates with water through an equilibrator. Therefore, the speed of measurements depends, among other factors (such as the size and type of the equilibrator), on the gas solubility. While $CO_2$ equilibrium was reached within minutes, $CH_4$ solubility caused equilibration to take much longer, particularly at high concentrations, such as those observed at Site 1.

This issue occurs not only when moving from low to high concentrations but also when shifting from high to low concentrations. Since our study examines the spatial variability of GHG concentrations, it was essential to ensure that each measurement truly reflected an equilibrium state. Therefore, to provide high-quality data, we allowed enough time—up to 45 minutes—for the system to reach complete equilibrium with the water.

The 45-minute timeframe is flexible and depends on how quickly we achieve equilibrium. While on site, we could monitor the changes in concentrations of both $CO_2$ and $CH_4$ on a laptop. Once $CH_4$ reached a plateau, we assumed equilibrium had been achieved. It could take up to 45 minutes at sites with high concentrations.

We added this precision at the L 121, which now reads:

"Equilibration between the seawater and gas phases was monitored in real time using a laptop connected to the IRGA. Equilibrium was considered reached when both $CO_2$ and $CH_4$ concentrations stabilized at a clear plateau. Depending on the concentration gradient between sites, this equilibration period could vary substantially. While $CO_2$ typically reached equilibrium within a few minutes, $CH_4$ required longer times (up to 45 minutes) to reach a stable plateau."

**l. 130 – 131:** The agreement between chamber-derived and calculated flux densities seems to be remarkably good. Since such chambers are known to suffer from artefacts under turbulent conditions, it would be good for the readers (potential users) to learn about the sampling conditions.

→ We modified the text as follows (L 141):

"Air-sea exchanges of $CO_2$ and $CH_4$ were measured using the accumulation chamber technique (Frankignoulle, 1988). The chamber consists of a polyethylene container (internal diameter: 34 cm, height: 14 cm, total volume = 11.4 L) connected in a closed loop to the IRGA with 1.5 m long Tygon tubing, which allows the chamber to move freely on the water surface."

Our main precaution was to avoid taking measurements when conditions were too rough to prevent water from entering the IRGA, which is a standard and essential step to protect the instrument.

**l. 155:** Here, it is not clear whether the standardised wind speeds were computed from the mean of wind speeds at 2 and 3.2 m height.

→ The line 172 now reads "Wind speed (in m s–1, METEK, uSonic-3 Scientific) and air temperature (in °C, Vaisala, HMP155) are measured at the 3.2 and 2 m height, respectively, above sea level at the newly established Integrated Carbon Observation System (ICOS) coastal site at TZS (ICOS code FI-Tvm; Fig. 1)."

We added the word 'respectively' to avoid confusion about the anemometer's height (3.2m), which is extrapolated to the standardized height of 10m.

**l. 157 and ff.:** I think it would be important, for the sake of clarity, to mention how well the in-situ atmospheric measurements fit/are comparable to the atmospheric measurements at this site. For instance, I noticed that the mean value reported for CO2 (406 ppm; l. 195), is considerably lower than values reported at PAL station at the time of sampling:
https://gml.noaa.gov/data/data.php?site=PAL¶meter_name=Carbon%2BDioxide&frequency=Discrete. The values publicly available for PAL oscillate between ca. 409 and 416 ppm for the same time period in 2023. Admittedly, these values are flagged as "preliminary", but even after taking the last year of calibrated (final) measurements and applying the global annual rate of increase as an index (see: https://gml.noaa.gov/ccgg/trends/), the mean atmospheric $xCO_2$ value would be ca. 414 ppm. I did not check for the other gases (and certainly would not expect to change much the overall distribution), but it still would be good to clearly state which values were used and why.

→ The average $CO_2$ (406 ppm) and $CH_4$ (2.05 ppm) atmospheric concentration reported from this survey is the average value from all the atmospheric measurements obtained while we were flushing the IRGA before the air-sea flux measurements (i.e., while establishing the baseline).

We added this precision in the text, L 147:

"Before conducting the chamber measurements, the IRGA baseline was established by flushing atmospheric air through the analyser. Based on these measurements, we estimated average atmospheric concentrations of 406 ppm for $CO_2$ and 2.05 ppm for $CH_4$ during the survey."

We are unsure that comparing with the PAL station will be most relevant, as the station is located in Lapland on land covered with Alpine tundra, about 200 km north of the Arctic Circle and 1200 km north of the Tvärminne Zoological Station (TZS). A more suitable reference might be the ICOS marine station Utø, in the Baltic Sea, approximately 200 km west of TZS. In Utø, the atmospheric concentrations of $CO_2$ and $CH_4$ range from 404 to 440 ppm and 1.98 to 2.14, respectively, over the same period as our study. The two datasets seem to align well, considering the variability observed in both cases.

The Tvärminne Zoological Station is also an ICOS station and measures $CO_2$ and $CH_4$ concentrations. However, the $CH_4$ sensor was not installed in 2023. Therefore, only $CO_2$ data are available on the ICOS data center, and for the same period of this study, reported atmospheric concentrations ranged from 398 to 442 ppm. Here as well, our reported values match quite well with the ICOS atmospheric data.

https://data.icos-cp.eu/portal/#%7B%22filterCategories%22:%7B%22project%22:%5B%22icos%22%5D,%22level%22:%5B1,2%5D,%22stationclass%22:%5B%22ICOS%22%5D%7D%7D

**l. 188:** This should read "coloured dissolved organic matter".

→ Thank you for the correction.

**l. 208 and ff.:** Some parts of Fig. 3 are addressed during the results section, while others are mentioned only later during the discussion. Since some of the results are referred to in Fig. 4, it is confusing for the reader. I kindly suggest considering to split Fig. 3 in separate figures that appear then in sections 3 and 4.

→ Thank you for the suggestion. We could split figure n°3 into two separate figures. The new figure n°3 could focus solely on the conservative behavior of both total alkalinity (TA) and dissolved inorganic carbon (DIC) with salinity (the current figure 3a from the manuscript). Figures examining the TA:DIC ratio in relation to surface water $pCO_2$, $CH_4$, and $N_2O$ concentrations (Figures 3b, c, and d in the current manuscript) could then be presented as a separate figure. As we discuss those figures later, as pointed out by the reviewer, this figure will then be numbered figure n°6.

While we understand the reasoning behind the request, we prefer to keep all figures related to TA and DIC in a single plot (current figure 3). This way, all information on those variables is consolidated in one place, reducing the total number of figures in the manuscript and preventing the reader from having to navigate between figures 3 and 6.

**l. 210 – 211:** I think "air-sea flux densities" would be a more adequate term here (instead of "exchanges").

→ We changed the title of the section to 'Air-sea GHG flux densities' and used similar wording in the section: "Measured air-sea $CO_2$ flux densities ranged from…", "… source of $CH_4$ to the atmosphere, with flux densities ranging from…

**l. 232 – 241:** My impression is that this analysis would be better suited for the discussion. Also, it would perhaps help the line of argumentation if variables which are mechanistically expected to be related are analysed. An example of this would be $N_2O$ and $CO_2$.

→ We believe the reviewer is requesting to move the section related to the statistical analysis from the Discussion to the Results section, as Truong An Nguyen (reviewer 1) did. This section has now been relocated to section 3.4.

All variables that are mechanistically expected to be related have been included in the analysis, as L 237 reads:

"The Kendall's $\tau$ coefficient has been calculated to investigate the correlation between surface water $pCO_2$, $CH_4$, and $N_2O$ concentration, as well as between physical and biogeochemical parameters and all three GHGs (Fig. 4)."

So this includes the correlation between $N_2O$ and $CO_2$, shown in Figure 4.

**l. 344 – 365:** I think comparing the radiative budget calculated for this system with global estimates (e.g. works by Rosentreter et al and Resplandy et al), would strengthen the arguments laid in this manuscript and emphasize its relevance.

→ Thank you. We followed the suggestion by modifying the conclusion to include work by Rosentreter et al. (2021) and Resplandy et al. (2024) to highlight the interest and need for this work, both at small and large scales. It would have been interesting to compare our GHG budget with the estimate from Resplandy et al. (2024). However, our budget is only valid for one month in late summer, while theirs covers an entire year, including spring and summer blooms—both events we did not observe during our survey. Therefore, a proper comparison would not be very relevant.

The end of the conclusion now reads:

"When translated into $CO_{2\text{-eq}}$, air–sea GHG fluxes were dominated by $CO_2$, while $CH_4$ and $N_2O$ contributed comparably but in different ways. $CH_4$ consistently acted as a source, whereas $N_2O$ partially offset the $CH_4$ release through uptake. This interaction highlights that the balance between production and consumption processes, especially within different seafloor habitats, is critical for understanding coastal contributions to the global carbon budget. There is a critical need to quantify $CH_4$ and $N_2O$ exchanges more accurately and to deepen our understanding of the environmental and management factors that control their production and consumption. This would help make global estimates less sensitive to statistical assumptions and reduce uncertainties in blue carbon potential estimates (Rosentreter et al. 2021a).

While coastal ecosystems are often recognized as $CO_2$ sinks, recent work shows that they are also significant sources of $CH_4$ and $N_2O$ that can offset a substantial portion of the climate benefit of taking up $CO_2$ (Rosentreter et al., 2021a; Roth et al., 2023; Resplandy et al., 2024). Given their extent, northern temperate coastal ecosystems represent a relevant but overlooked source of GHGs, with the potential to amplify the global ocean carbon budget by increasing net greenhouse gas emissions to the atmosphere. Our results highlight the urgent need for research that integrates GHG with biodiversity, benthic–pelagic interactions, and microbial processes, and that resolves temporal variability across seasonal cycles. Such knowledge is essential to improve predictions of how coastal ecosystems will mediate carbon–climate feedback under future environmental change."

**l. 392:** It is not clear what is meant by "amplify" the global ocean carbon budget.

→ We changed the sentence as follows: "Given their extent, northern temperate coastal ecosystems represent a relevant but overlooked source of GHGs, with the potential to amplify the global ocean carbon budget by increasing net greenhouse gas emissions to the atmosphere."

Kind regards,
Damian L. Arévalo-Martínez

**Reviewer n°3**

Overview:
The manuscript „Spatial heterogeneity of GHG dynamics across an estuarine ecosystem" presents a nice data set of GHG data and related variables in an estuarine ecosystem in the northern Baltic Sea. The data set presents a spatial distribution within the onset of the of the salinity gradient, however, it represents just the late summer period. Nevertheless, the data set included next to the CO2, Methan and N2O as well. The main findings are that the estuarine system is heterogeneous, sinks and sources were found for CO2 and N2O, but methane were supersaturated in all samples and were emitted to the atmosphere.

The manuscript is well written and relatively easy to follow, even if you are not familiar with the study site. Although it is known that the coastal waters and estuaries are important for the GHG balance, comprehensive data set with all three most important GHG are still rare, and they are needed to understand these quite heterogeneous contributions.

Major comments:

If next to CO2 and Methane also N2O is included to understand the interaction between these three GHGs, it would be good to introduce next to the Carbon cycle also the Nitrogen Cycle and the interaction between them. At some points the mentioning of N2O not good connected.

→ We modified the introduction. We can now read starting at L 65:

"N$_2$O is primarily produced via nitrification (ammonia oxidation) and denitrification (nitrate reduction) in sediments and water columns and is controlled by the availability of dissolved inorganic nitrogen and oxygen (Bange 2006). Eutrophication and hypoxia, often driven by excess nutrient and organic matter inputs, have been shown to promote N$_2$O emissions (Murray et al., 2015; Brase et al., 2017). Coastal ecosystems are recognized as significant sources of N$_2$O to the atmosphere (Bange, 2006; Cheung et al., 2025; Resplandy et al., 2024), where denitrification, especially in sediments and on particles, often dominates N$_2$O production, even in well-oxygenated waters (Wan et al., 2023). However, coastal estimates remain uncertain due to sparse measurements and high spatial heterogeneity (Wan et al., 2023)."

One important variable to understand N2O production and emissions is next to the oxygen supply the availability of nitrate and the other DIN, as the authors mentioned in the discussion. Can you present also at least nitrate concentrations and include them in the correlation matrix?

→ Unfortunately, we did not measure nitrate and other DIN during our sampling. Some measurements were taken in the study area during similar times, but our sampling was not conducted at the same time and/or exact location, making it difficult to use them to improve the interpretation of the N$_2$O data collected. Maybe our sentence was unclear; we modified it as follows:

L 257:

"Aalto et al. (2021) reported that higher N$_2$O concentrations were associated with higher nitrate concentrations and inputs of allochthonous carbon, while lower N$_2$O concentrations were associated with efficient internal recycling of N."

L 355:

"Spatial variability of N$_2$O mirrors the findings of Aalto et al. (2021), who linked higher N$_2$O concentrations near the Karjaanjoki River to higher nitrate inputs and allochthonous carbon inputs."

Somewhere in the manuscript the authors must explain what the potential sinks for N2O are and present an explanation for the undersaturation of the N2O. Do you assume that N2O is consumed under anaerobic conditions in the sediments? Are the also processes in the water column which consumed N2O.

→ The lack of nitrate measurements associated with our observations makes it difficult to explain the observed trend, but a study by Aalto et al. (2021) from the same area, along the same salinity gradient, suggested that sites located in the 'offshore archipelago,' where autochthonous (locally produced, e.g., phytoplankton-derived) organic matter is more common and nitrate concentrations are lower, promote DNRA as the main pathway for recycling nitrate to ammonium without producing N$_2$O. This process acts as a sink by suppressing N$_2$O formation. They also suggest that complete denitrification occurred in environments with a high organic carbon-to-nitrate ratio, such as those with abundant autochthonous organic matter.

We modified our manuscript and can now read from L 353:

"Spatial variability of $N_2O$ mirrors the findings of Aalto et al. (2021), who linked higher $N_2O$ concentrations near the Karjaanjoki River to higher nitrate inputs and allochthonous carbon inputs. They suggested that the ratio between nitrate and autochthonous organic carbon controls the balance between N-removing denitrification and N-recycling through Dissimilatory Nitrate Reduction to Ammonium (DNRA), as well as the end-product of denitrification (Aalto et al., 2021). Within the archipelago, where riverine influence is limited, DNRA can produce significant amounts of bioavailable ammonium, enhancing nitrogen recycling between sediments and surface water, especially in summer, when autochthonous biomass production and sedimentation are highest."

The aim of the study which is formulated at the end of the introduction seems to be a bit too ambiguous. I would suggest orientating the aims a bit more on the structure of the discussion section.

→ We rephrased the last chapter of the introduction (L 77). It now reads:

"We investigate the dynamics of $CO_2$, $CH_4$, and $N_2O$ (GHGs) along a salinity gradient and across contrasting coastal habitats within an estuary. We combined detailed field measurements of surface seawater physical and biogeochemical properties with both in situ measurements and calculated estimates of air–sea GHG exchange. This approach allowed us to capture spatial variability and discuss the roles of physical drivers and biological processes in the observed changes in GHG concentrations. In doing so, we aim to estimate the contribution of coastal ecosystems to GHG emissions and provide additional data that may be useful in improving global estimates of GHG emissions from the coastal ocean."

The explanation of the importance of the sediments for the GHG production, without having direct measurements comes a bit out of the blue and contradicts the hypotheses that the GHGs are mainly just mixed from the river inputs.

→ We have tuned down the importance of the sediments for GHG production, as we don't have direct measurements. In the new section 4.3 Exposed and semi-sheltered vs sheltered sites, we can read at L 338:

"Freshwater mixing with seawater appears to control the concentration of all three GHGs at exposed and semi-sheltered sites, as much of the variability in surface seawater concentrations can be explained by salinity. In sheltered sites of the archipelago, deviations from the salinity-driven pattern indicate the influence of additional processes."

Further down, L 349, we can read:

"Anoxic degradation of organic-rich sediments (methanogenesis), exacerbated by warm late-summer waters (Roth et al., 2022; Yvon-Durocher et al., 2014), combined with short water residence times that limit oxidation in both sediment and the overlying water column (Reeburgh, 2007), creates favourable conditions for $CH_4$ production. Due to the sheltered nature of the sites and limited water exchange, the produced $CH_4$ can accumulate. Similarly, enhanced respiration of organic carbon in shallow ecosystems elevates $CO_2$ concentrations (Humborg et al., 2019)."

Some specific / minor comments:

Abstract: quite too long and can be more focused.

→ We tried to shorten it, taking into consideration comments from other reviewers.

L21: N2O is missing

→ We are not sure about what the reviewer is referring to.
L 21 reads: "$CH_4$ concentrations were consistently supersaturated (19 to 469 nmol $L^{-1}$) compared to the atmosphere, resulting in a net source to the atmosphere from 0.014 to 1.39 mmol $m^{-2}$ $d^{-1}$."
This sentence is not related to $N_2O$, and the previous sentence clearly refers to both $CO_2$ and $N_2O$.

L 28: Is Methane always a source and N2O a sink?

→ The sentence has been changed as follows: "The overall budget of air–sea GHG exchanges was dominated by $CO_2$ fluxes, with $CH_4$ consistently acting as a source, and $N_2O$ alternating between source and sink."

L64. There is some literature that N2O production and Oxygen depletion correlated, e.g. in the Elbe Estuary

→ We modified this section; see earlier response, where we mentioned studies on $N_2O$ dynamics in the Elbe Estuary.

L 88: What are soft-sediment habitats

→ The sentence now reads (L 93):

> "Sampling sites (N = 21) were selected to encompass a wide range of soft-sediment habitats (e.g., including both vegetated and non-vegetated sediments with grain size ranging from coarse sand to clay, silt, and mud) and to represent a spatial gradient (50 km) from the outer to innermost archipelago."

L100. What was the partial sampling strategy? From fresh to salty? Each Stations just once, how much on one day….

→ We added the following precision (L 110):

> "Sites were visited once during the study period, with sampling conducted every two to three days, depending on weather conditions and boat availability."

L231: Did you measure DIN (Nitrate, …)

→ Unfortunately, we did not measure nitrate and other DIN during our sampling. Some measurements were taken in the study area during similar times, but our sampling was not conducted at the same time and/or exact location, making it difficult to use them to improve the interpretation of the $N_2O$ data collected.

L 306 ff. Here comes the main concrete results of the manuscript that can be one new chapter

→ We have not split section "4. Discussion" into four subsections:
   4.1. Salinity gradient
   4.2 Biological driver
   4.3 Exposed and semi-sheltered *vs* sheltered sites (the new chapter as requested)
   4.4 Air-sea flux densities